# Towards Identifiability of Hierarchical Temporal Causal Representation Learning

**Zijian Li**[1,2*]    **Minghao Fu**[4,2,*]    **Junxian Huang**[3]    **Yifan Shen**[2]    **Ruichu Cai**[3,†]
**Yuewen Sun**[1,2]    **Guangyi Chen**[1,2]    **Kun Zhang**[1,2,†]

[1]Carnegie Mellon University
[2]Mohamed bin Zayed University of Artificial Intelligence
[3]Guangdong University of Technology
[4] University of California San Diego

## Abstract

Modeling hierarchical latent dynamics behind time series data is critical for capturing temporal dependencies across multiple levels of abstraction in real-world tasks. However, existing temporal causal representation learning methods fail to capture such dynamics, as they fail to recover the joint distribution of hierarchical latent variables from *single-timestep observed variables*. Interestingly, we find that the joint distribution of hierarchical latent variables can be uniquely determined using three conditionally independent observations. Building on this insight, we propose a **C**ausally **Hi**erarchical **L**atent **D**ynamic (**CHiLD**) identification framework. Our approach first employs *temporal contextual observed variables* to identify the joint distribution of multi-layer latent variables. Sequentially, we exploit the natural sparsity of the hierarchical structure among latent variables to identify latent variables within each layer. Guided by the theoretical results, we develop a time series generative model grounded in variational inference. This model incorporates a contextual encoder to reconstruct multi-layer latent variables and normalize flow-based hierarchical prior networks to impose the independent noise condition of hierarchical latent dynamics. Empirical evaluations on both synthetic and real-world datasets validate our theoretical claims and demonstrate the effectiveness of **CHiLD** in modeling hierarchical latent dynamics. [3]

## 1 Introduction

Hierarchical temporal structures are pervasive in time series data such as weather records and stock prices. These structures arise from latent processes operating at multi-level abstraction—for example, seasonal, monthly, and daily variations in climate data. Understanding the hierarchical latent dynamics underlying time series remains a fundamental challenge [68, 4, 70] and has received growing attention. The core of this problem is the identification of latent processes evolving across different levels of abstraction from observed data.

To identify the temporal latent process, several methods have considered Independent Component Analysis (ICA) [30, 8, 27] to identify latent variables. To extend to nonlinear scenarios, researchers use various assumptions, such as sufficient changes [37, 72, 47, 77], to ensure the independent variation of latent variables. Specifically, some approaches leverage auxiliary variables [18, 19, 28, 29, 37, 31] to achieve strong identifiability of latent variables. In the context of time-series data, others utilize historical observations as surrogates for historical latent variables to induce sufficient changes [73, 29].

---

[*]Equal contribution.
[†]Corresponding authors.
[3]https://github.com/MinghaoFu/CHiLD

39th Conference on Neural Information Processing Systems (NeurIPS 2025).

Recent advances have further tackled the challenge of identifying latent dynamics with instantaneous dependencies under assumptions like interventions [54], grouping observations [57], and the sparse causal influence [48]. Please refer to more discussion of the related works and real-world implications of hierarchical latent dynamics in Appendix E and C, respectively.

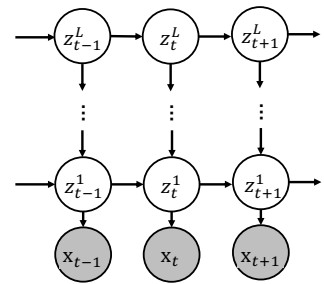

Despite recent advances, these methods predominantly assume a single layer of latent variables and will fail to identify hierarchical latent dynamics. As a result, under a hierarchical structure, using historical observation as a surrogate to leverage the sufficient changes condition is not applicable anymore since the high-level variables will introduce noise. Consequently, methods under the single-layer assumption [28, 74] face challenges in recovering the joint distribution of hierarchical latent variables from single-timestep observations, particularly when the observations are derived from lower-level variables. This limitation prevents these methods from using historical observations as surrogates to meet the sufficient changes condition, resulting in suboptimal identifiability.

Figure 1: Illustration of data generation process with hierarchical temporal dynamics that consists of $L$-layer latent variables. The observed variables $\mathbf{x}_t$ are generated by $\mathbf{x}_t = \mathbf{g}(\mathbf{z}_t^1, \epsilon_t^0)$, where $\mathbf{g}$ and $\epsilon_t^0$ denote the nonlinear mixing function and noise, respectively. And $\mathbf{z}_t^l$ are influenced by its time-delayed and hierarchical parents $\mathbf{z}_{t-1}^l$ and $\mathbf{z}_t^{l+1}, l \leq L-1$, respectively.

The point above highlights the urgent need for temporal causal representation learning with hierarchical latent dynamics is to recover the joint distribution of hierarchical latent variables. Interestingly, we find that the distribution of latent variables can be uniquely determined with the help of three temporally adjacent observations [23, 25]. Furthermore, we demonstrate that even with hierarchical structures in the latent processes, the joint distribution of multi-layer latent variables can be identified by incorporating additional temporal contextual observations. Therefore, we can derive a hierarchical dynamic process underlying the observed data via the natural sparsity of the hierarchical latent structure, facilitating time series generation grounded in a causal understanding of the process.

Based on this insight, we therefore establish a **C**ausally **Hi**erarchical **L**atent **D**ynamic (**CHiLD**) identification framework by harnessing the temporal contextual observations and natural sparsity of the hierarchical structure among latent variables. Additionally, to connect the theoretical results with a practical algorithm, we develop a time series generative model based on a variational autoencoder with a contextual encoder and normalizing flow-based hierarchical prior networks. Specifically, to identify the joint distribution of multi-layer latent variables, the contextual encoder transforms the historical, current, and future observations into the hierarchical latent variables at the current timestamp. Moreover, to enforce the independent noise condition of hierarchical latent dynamics, the normalizing flow-based hierarchical prior networks determine the prior distribution of hierarchical latent variables. These two specialized modules together boost the identification of hierarchical temporal causal representation learning. We validate our approach through simulation experiments to demonstrate identifiability and evaluate its performance on eleven time-series generation benchmarks for both generation quality and controllability. The impressive quantitative and qualitative results underscore the effectiveness of our method.

## 2 Preliminaries

### 2.1 Hierarchical Data Generation Process

We begin with the time series generation process with a hierarchical latent causal process as shown in Figure 1 [4]. To facilitate clarity, we adopt the terminology widely used in ICA literature like observed variables $\mathbf{x}_t$, latent variables $\mathbf{z}_t$, and the mixing function $\mathbf{g}$. In the context of hierarchical latent dynamics, we let $\mathbf{z}_t = \{\mathbf{z}_t^1, \cdots, \mathbf{z}_t^l, \cdots, \mathbf{z}_t^L\}$, where $\mathbf{z}_t^l \in \mathbb{R}^n$ denotes the $l$-th layer latent variables. Suppose that we have observed variables with discrete timestamps $\boldsymbol{X} = \{\mathbf{x}_1, \cdots, \mathbf{x}_t, \cdots, \mathbf{x}_T\}$, where $\mathbf{x}_t \in \mathbb{R}^n$ is only generated from first layer latent variables $\mathbf{z}_t^1 \in \mathcal{Z} \subseteq \mathbb{R}^n$ (low-level latent

---

[4]Illustration shows a first-order Markov structure, our method can extend to higher-order dependencies.

variables) via a nonlinear mixing function $\mathbf{g}$ and independent noise $\epsilon_t^0$. Mathematically, we have:

$$\mathbf{x}_t = \mathbf{g}(\mathbf{z}_t^1, \epsilon_t^0). \tag{1}$$

For the $l$-th layer, $i$-th dimension latent variable at $t$ timestamp $z_{t,i}^l, l \in \{1, \cdots, L-1\}$ is generated via the latent causal procedure, which is assumed to be related to time-delayed parent $\mathrm{Pa}_d(z_{t,i}^l)$ and the hierarchical parents $\mathrm{Pa}_h(z_{t,i}^l)$, respectively. Formally, it can be written via a structural equation model as follows

$$z_{t,i}^l = f_i^l(\mathrm{Pa}_d(z_{t,i}^l), \mathrm{Pa}_h(z_{t,i}^l), \epsilon_{t,i}^l), \quad \text{with} \quad \epsilon_{t,i}^l \sim p_{\epsilon_{t,i}^l}, \tag{2}$$

where $\epsilon_{t,i}^l$ is the temporally and spatially independent noise sampled from $p_{\epsilon_{t,i}^l}$. And the $L$-th layer latent variables $\mathbf{z}_t^L$ is only related to the time-delayed parent $\mathrm{Pa}_d(z_{t,i}^L)$, which is formalized as follows:

$$z_{t,i}^L = f_i^L(\mathrm{Pa}_d(z_{t,i}^L), \epsilon_{t,i}^L), \quad \text{with} \quad \epsilon_{t,i}^L \sim p_{\epsilon_{t,i}^L}. \tag{3}$$

Note that the hierarchical structure among latent variables of data generation process in Figure 1 and Equation (1)-(3) is naturally sparse, since there are no direct causal relations between $\mathbf{z}_t^l$ and $\mathbf{z}_{t-1}^{l+1}$, meaning that $\mathbf{z}_t^l$ is independent of $\mathbf{z}_{t-1}^{l+1}$ conditional on other latent variables.

Based on the generation process above, we further define the block-wise identifiability and component-wise identifiability of latent variables in Definition 1 and 2, respectively.

**Definition 1** (**Block-wise Identifiability of Latent Variables $\mathbf{z}_t^l$** [69]). *The block-wise identifiability of $\mathbf{z}_t^l \in \mathbb{R}^n$ means that for ground-truth $\mathbf{z}_t^l$, there exists $\hat{\mathbf{z}}_t^l \in \mathbb{R}^n$ and an invertible function $h : \mathbb{R}^n \to \mathbb{R}^n$, such that $\mathbf{z}_t^l = h(\hat{\mathbf{z}}_t^l)$.*

**Definition 2** (**Component-wise Identifiability of Latent Variables $z_{t,i}^l$** [42]). *The component-wise identifiability of $\mathbf{z}_t^l \in \mathbb{R}^n$ is that for each $\mathbf{z}_{t,i}^l, i \in \{1, \cdots, n\}$, there exists a corresponding estimated component $\hat{\mathbf{z}}_{t,j}^l, j \in \{1, \cdots, n\}$ and an invertible function $h_i^l : \mathbb{R} \to \mathbb{R}$, such that $z_{t,i}^l = h_i^l(\hat{z}_{t,j}^l)$.*

### 2.2 Why previous theoretical results can hardly identify the hierarchical latent variables?

Based on the data generation process described in Equations (1)-(3), we provide a detailed explanation of why previous methods fail to identify hierarchical latent variables. First, the previous methods for temporal causal representation learning [28, 29, 73, 74] usually assume that the observed variables $\mathbf{x}_t$ are generated from a single-layer latent variables $\mathbf{z}_t$ via an invertible and deterministic mixing function $\mathbf{g}$, e.g., $\mathbf{x}_t = \mathbf{g}(\mathbf{z}_t)$, implying that we can recover the distribution of latent variables from observed variables by matching the marginal distribution of observed variables. Sequentially, these methods can leverage the historical observation $\mathbf{x}_{t-1}$ as a surrogate for $\mathbf{z}_{t-1}$ to meet the sufficient change condition and further achieve identifiability.

However, when the data generation process follows a hierarchical latent dynamics (we suppose there are two latent layers for convenience), since the low-level latent variables, e.g., $\mathbf{z}_t^1$ are generated from the high-level latent variables e.g., $\mathbf{z}_t^2$, and noise $\epsilon_t^1$, we cannot recover the joint distribution of $\mathbf{z}_t = \{\mathbf{z}_t^1, \mathbf{z}_t^2\}$ due to the additional noise $\epsilon_t^1$. Therefore, previous methods cannot consider $\mathbf{x}_{t-1}$ as the surrogate of $\mathbf{z}_{t-1}$, and hence cannot achieve identification under the hierarchical latent dynamics.

## 3 Identifiability of Hierarchical Latent Variables

In this section, we show how to identify the hierarchical latent dynamics under temporal contextual observations. Specifically, we first establish the block-wise identifiability results of multi-layer latent variables $\mathbf{z}_t = \{\mathbf{z}_t^1, \cdots, \mathbf{z}_t^L\}$ with $2L + 1$ adjacent observed variables (Theorem 1). Sequentially, we leverage the estimated multi-layer latent variables $\hat{\mathbf{z}}_t$ as a surrogate of the true variables $\mathbf{z}_t$ to meet the sufficient change condition. Then we leverage the connection between natural sparsity of the hierarchical structure among latent variables and cross derivative [52] to achieve block-wise identifiability for latent variables within each layer (Theorem 2). Finally, we further exploit the previous results [74] to achieve component-wise identifiability of latent variables $z_{t,i}^l$ (Lemma 1).

## 3.1 Identifiability of Multi-layer latent variables $\mathbf{z}_t$

In this subsection, we show the identifiability results of multi-layer latent variables $\mathbf{z}_t$. For a better explanation, we first provide the definition of the linear operator [23, 25] as follows.

**Definition 3** (**Linear Operator** [23, 11]). *Consider two random variables $\mathbf{a}$ and $\mathbf{b}$ with support $\mathcal{A}$ and $\mathcal{B}$, the linear operator $L_{\mathbf{b}|\mathbf{a}}$ is defined as a mapping from a probability function $p_{\mathbf{a}}$ in some function space $\mathcal{F}(\mathcal{A})$ onto the probability function $p_{\mathbf{b}} = L_{\mathbf{b}|\mathbf{a}} \circ p_{\mathbf{a}}$ in some function space $\mathcal{F}(\mathcal{B})$,*

$$\mathcal{F}(\mathcal{A}) \to \mathcal{F}(\mathcal{B}) : p_{\mathbf{b}} = L_{\mathbf{b}|\mathbf{a}} \circ p_{\mathbf{a}} = \int_{\mathcal{A}} p_{\mathbf{b}|\mathbf{a}}(\cdot|\mathbf{a})p_{\mathbf{a}}(\mathbf{a})d\mathbf{a}. \tag{4}$$

Based on the definition of linear operator, we show that the multi-layer latent variables $\mathbf{z}_t = \{\mathbf{z}_t^1, \cdots, \mathbf{z}_t^L\}$ can be block-wise identifiability as follows.

**Theorem 1.** *(**Block-wise Identifiability of Latent Variables** $\mathbf{z}_t$ **in Hierarchical Latent Process.**) Suppose the observed and L-layer latent variables follow the data generation process in Figure 1. By matching the true joint distribution of $2L + 1$ number of adjacent observed variables, i.e., $\{\mathbf{x}_{t-L}, \cdots, \mathbf{x}_t, \cdots, \mathbf{x}_{t+L}\}$, we further make the following assumptions:*

*(i) The joint distribution of $\mathbf{x}, \mathbf{z}$, and their marginal and conditional densities are bounded and continuous.*

*(ii) The linear operators $L_{\mathbf{x}_{t+1}, \cdots, \mathbf{x}_{t+L}|\mathbf{z}_t}$ and $L_{\mathbf{x}_{t-L}, \cdots, \mathbf{x}_{t-1}|\mathbf{x}_{t+1}, \cdots, \mathbf{x}_{t+L}}$ are injective for bounded function space.*

*(iii) For all $\mathbf{z}_t, \mathbf{z}_t' \in \mathcal{Z}_t$ with $\mathbf{z}_t \neq \mathbf{z}_t'$, the set $\{\mathbf{x}_t : p(\mathbf{x}_t|\mathbf{z}_t) \neq p(\mathbf{x}_t|\mathbf{z}_t')\}$ has positive probability.*

*Suppose that we have learned $(\hat{g}, \hat{f}, p_{\hat{\epsilon}})$ to achieve Equations (1) and (2), then the latent variables $\mathbf{z}_t = \{\mathbf{z}_t^1, \cdots, \mathbf{z}_t^L\}$ are block-wise identifiable.*

**Intuition and Proof Sketch.** Compared with previous works that leverage spectral decomposition to achieve identifiability [23, 14], we do not require the monotonicity and normalization assumption [23] (see Appendix B.4) and quantitatively formulate the relations between required observations and the latent layers. Intuitively, Theorem 1 shows that at least $2L + 1$ adjacent observations are the minimal information for the identification of $\mathbf{z}_t$ under the data generation process described in Equations (1)-(3). This is because of the injectivity of the two operators, where an operator that maps some functional space onto lower dimensions cannot be injective by Definition 3. The proof can be summarized in the two steps. First, by using the fact that $\{\mathbf{x}_{t-L}, \cdots, \mathbf{x}_{t-1}\}$, $\{\mathbf{x}_t\}$, and $\{\mathbf{x}_{t+1}, \cdots, \mathbf{x}_{t+L}\}$ are conditionally independent given $\mathbf{z}_t$, we can construct an eigenvalue-eigenfunction decomposition regarding the integral operator. Next, by leveraging the relationship between the uniqueness of spectral decomposition (Theorem XV.4.3.5 [11]), latent variables are block-wise identifiable when the marginal distribution of observed variables is matched.

**Implication.** Theorem 1 demonstrates that temporal contextual observations can be used to identify the joint distribution of multi-layer latent variables, even when additional noises are introduced into the generation process from high-level latent variables to observations. As an additional benefit, the same identification result can be achieved even when the mixing process from latent to observed variables is non-deterministic and involves independent noise.

**Discussion on Assumptions.** To clarify our theoretical results, we provide an explanation of these assumptions and their relevance to real-world scenarios. **1)** The assumption of the bounded and continuous conditional densities is standard in the literature on the identification of latent variables under measurement error [23, 25], meaning that the observed and latent variables are bounded and change continuously. For example, in the financial markets example, the stock prices (observed variables) and the interest rates (latent variables) usually change continuously. And their values are usually within a certain reasonable range. **2)** The injective linear operator assumption is also used in the literature on the identification of latent variables under measurement error [14]. (More examples of this condition can be found in Appendix B.3). Intuitively, an injective operator $L_{b|a}$ implies that there is enough variation in the density of $b$ for different distributions of $a$. Using the same financial markets example, the injectivity of $L_{\mathbf{x}_{t+1}, \cdots, \mathbf{x}_{t+L}|\mathbf{z}_t}$ means that the profitability (i.e., $\mathbf{z}_t$) of a company has sufficient influence on the stock prices of that company (i.e., $\mathbf{x}_{t+1}, \cdots, \mathbf{x}_{t+L}$). And $L_{\mathbf{x}_{t-L}, \cdots, \mathbf{x}_{t-1}|\mathbf{x}_{t+1}, \cdots, \mathbf{x}_{t+L}}$ means that the stock price of a company has a significant impact on its future stock price. Therefore, this assumption is easy to meet in practice. **3)** The third assumption means that the density of $\mathbf{x}_t$ can be different when the values of $\mathbf{z}_t$ are different, which is also easy to meet. For example, under different interest rates, the variance of the stock price is usually different.

## 3.2  Identifiabiliy of Single-layer Latent Variables $\mathbf{z}_t^l$

**Theorem 2.** (*Block-wise Identifiability of Latent Variables $\mathbf{z}_t^l$ in any $l$-th Layer.*) *For a series of observed variables $\mathbf{x}_t \in \mathbb{R}^n$ and estimated latent variables $\hat{\mathbf{z}}_t^l \in \mathbb{R}^n$ with the corresponding process $\hat{f}_i, \hat{P}(\hat{\epsilon}), \hat{\mathbf{g}}$, suppose that the process subject to observational equivalence $\mathbf{x}_t = \hat{\mathbf{g}}(\hat{\mathbf{z}}_t^1, \hat{\epsilon}_t^0)$. We let $\mathbf{c}_t \triangleq \{\mathbf{z}_{t-1}, \mathbf{z}_t\} \in \mathbb{R}^{2 \times L \times n}$ and that $\mathcal{M}_{\mathbf{c}_t}$ be the variable set of two consecutive timestamps and the corresponding Markov network respectively, and further employ following assumptions:*

*(i) (Smooth and Positive Density [74, 42]): The conditional probability function of the latent variables $\mathbf{c}_t$ is smooth and positive.*

*(ii) (Sufficient Variability) [48]: Denote $|\mathcal{M}_{\mathbf{c}_t}|$ as the number of edges in Markov network $\mathcal{M}_{\mathbf{c}_t}$. Let*

$$w(m) = \left( \frac{\partial^3 \log p(\mathbf{c}_t | \mathbf{z}_{t-2})}{\partial c_{t,1}^2 \partial z_{t-2,m}}, \cdots, \frac{\partial^3 \log p(\mathbf{c}_t | \mathbf{z}_{t-2})}{\partial c_{t,2n}^2 \partial z_{t-2,m}} \right) \oplus \left( \frac{\partial^2 \log p(\mathbf{c}_t | \mathbf{z}_{t-2})}{\partial c_{t,1} \partial z_{t-2,m}}, \cdots, \frac{\partial^2 \log p(\mathbf{c}_t | \mathbf{z}_{t-2})}{\partial c_{t,2n} \partial z_{t-2,m}} \right)$$
$$\oplus \left( \frac{\partial^3 \log p(\mathbf{c}_t | \mathbf{z}_{t-2})}{\partial c_{t,i} \partial c_{t,j} \partial z_{t-2,m}} \right)_{(i,j) \in \mathcal{E}(\mathcal{M}_{\mathbf{c}_t})}, \tag{5}$$

*where $\oplus$ denotes concatenation operation and $(i,j) \in \mathcal{E}(\mathcal{M}_{\mathbf{c}_t})$ denotes all pairwise indice such that $c_{t,i}, c_{t,j}$ are adjacent in $\mathcal{M}_{\mathbf{c}_t}$. For $m \in [1, \cdots, n]$, there exist $4n + |\mathcal{M}_{\mathbf{c}_t}|$ different values of $\mathbf{z}_{t-2,m}$, such that the $4n + |\mathcal{M}_{\mathbf{c}_t}|$ values of vector functions $w(m)$ are linearly independent.*

*Then for latent variables $\mathbf{z}_t^l$ at the $l$-th layer, $\mathbf{z}_t^l$ is block-wise identifiable without permutation, i.e., there exists $\hat{\mathbf{z}}_t^l$ and an invertible function $h^l : \mathbb{R}^n \to \mathbb{R}^n$, such that $\mathbf{z}_t^l = h^l(\hat{\mathbf{z}}_t^l)$.*

**Intuition and Proof Sketch.** Proof can be found in Appendix A3. This proof is built upon the results of Theorem 1, since we cannot achieve the joint distribution of latent variables in the hierarchical structure. By considering latent variables $\mathbf{c}_t = \{\mathbf{z}_{t-1}, \mathbf{z}_t\}$ from two adjacent timestamps, the cross-layer latent variables are conditionally independent, e.g, $\mathbf{z}_{t,i}^l \perp\!\!\!\perp \mathbf{z}_{t-1,j}^{l+1} | \mathbf{c}_t \backslash \{\mathbf{z}_{t,i}^l, \mathbf{z}_{t-1,j}^{l+1}\}, \forall i, j \in \{1, \cdots, n\}$. And such conditional independence implies $\frac{\partial^2 \log p(\mathbf{c}_t)}{\partial \mathbf{z}_{t,i}^l \partial \mathbf{z}_{t-1,j}^{l+1}} = 0$ by leveraging the connection between conditional independence and cross derivatives [52]. Moreover, since we have achieved the block-wise identifiability of $\mathbf{z}_t$, we can leverage the contextual observation as the surrogates of historical latent variables and further introduce the sufficient variability assumption [48]. As a result, we can construct a full-rank linear system with the unique solution. By further exploiting the hierarchical temporal structure of latent variables, we can show that the estimated $\hat{\mathbf{z}}_t^l$ can be block-wise identifiable without permutation, i.e, there exists an invertible function $h^l : \mathbb{R}^n \to \mathbb{R}^n$, such that $\mathbf{z}_t^l = h^l(\hat{\mathbf{z}}_t^l)$.

**Compared with Previous Theoretical Results.** Although both [48] and Theorem 2 use the sparsity influence of latent dynamics to achieve identifiability, yet our contribution goes substantially further. First, IDOL [48] assumes an invertible mixing procedure, which begins from matching the marginal distribution of $\mathbf{x}_t$. While our method allows that partial latent variables contribute to the observations with a noise-contaminated mixing procedure, so it is built on the block-wise identification from Theorem 1. Second, IDOL shows the component-wise identifiability with permutation. However, our CHiLD shows the block-wise identifiable without permutation (please find it at Lemma A2 of Appendix B.5). This result is stronger than that of IDOL. We further show that each component within each layer $z_{t,i}^l$ is component-wise identifiable.

**Discussion on Assumptions.** We further explain the assumptions and their real-world implications. First, the smooth and positive density assumption is a commonly adopted assumption in nonlinear ICA studies [37, 74], implying that the latent variables can change continuously based on historical information. This assumption can be easily met in real-world scenarios like financial markets. For example, the exchange rates usually change smoothly over time.

Second, the sufficient variability assumption is widely employed in existing results on the identifiability of temporally causal representation learning [73, 74, 5, 48], reflecting the changeability of latent variables. Take stock price forecasting as an example, the latent variables may represent quantitative factors such as the price-to-earnings ratio, trading volume spread at different time steps. The linear independence of the latent variables implies that changes in one factor, such as the price-to-earnings ratio, cannot be linearly represented by trading volume. Moreover, while the sufficiency assumption forms the foundation of identifiability theory, it is not overly restrictive in practical applications.

Table 1: Attributes of causal representation learning theories. A check denotes that a method has an attribute or can be applied to a setting, whereas a cross denotes the opposite.

| Method | Hierarchical Structure | Noisy Mixing Procedure | Instantaneous Dependency | Time Series Data | Stationarity |
|---|---|---|---|---|---|
| PCL [29] | ✗ | ✗ | ✗ | ✓ | ✓ |
| TDRL [74] | ✗ | ✗ | ✗ | ✓ | ✓ |
| CaRiNG [5] | ✗ | ✗ | ✗ | ✓ | ✓ |
| General [77] | ✗ | ✗ | ✓ | ✗ | ✗ |
| IDOL [48] | ✗ | ✗ | ✓ | ✓ | ✓ |
| CHiLD | ✓ | ✓ | ✓ | ✓ | ✓ |

**Lemma 1.** (*Component-wise Identifiability of Latent Variables* $\mathbf{z}_{t,i}^l$ *in any* $l$-*th Layer* [74, 42])
*For a series of observed variables* $\mathbf{x}_t \in \mathbb{R}^n$ *and estimated latent variables* $\hat{\mathbf{z}}_t \in \mathbb{R}^{n \times L}$ *with the corresponding process* $\hat{f}_i, \hat{P}(\hat{\epsilon}), \hat{\mathbf{g}}$, *suppose that the process subject to observational equivalence* $\mathbf{x}_t = \hat{\mathbf{g}}(\hat{\mathbf{z}}_t, \hat{\epsilon}_t^0)$. *Besides the conditional independence assumption, i.e.,* $\log p(\mathbf{z}_t^l | \mathbf{z}_{t-1}^l, \mathbf{z}_t^{l+1}) = \sum_{i=1}^n \log p(z_{t,i}^l | \mathbf{z}_{t-1}^l, \mathbf{z}_t^{l+1})$, *we further assume the sufficient variability for each latent component. For any* $\mathbf{z}_t^l \in \mathcal{Z}_t^l \subseteq \mathbb{R}^n$ *and* $\hat{\mathbf{u}} = \{\hat{\mathbf{z}}_{t-1}^l, \hat{\mathbf{z}}_t^{l+1}\}$ *there exist* $2n+1$ *different values of* $\hat{\mathbf{u}}$, $m = 0, \cdots, 2n$, *such that these* $2n$ *vectors* $\mathbf{v}_{l,m}$ - $\mathbf{v}_{l,0}$ *are linearly independent, where* $\mathbf{v}_{l,m}$ *is defined as:*

$$\mathbf{v}_{l,m} = \Big( \frac{\partial^2 \ln P(z_{t,1}^l | \hat{\mathbf{u}})}{(\partial z_{t,1}^l)^2}, \cdots, \frac{\partial^2 \ln P(z_{t,n}^l | \hat{\mathbf{u}})}{(\partial z_{t,n}^l)^2}, \frac{\partial \ln P(z_{t,1}^l | \hat{\mathbf{u}})}{\partial z_{t,1}^l}, \cdots, \frac{\partial \ln P(z_{t,n}^l | \hat{\mathbf{u}})}{\partial z_{t,n}^l} \Big). \quad (6)$$

*Then for* $i$-*th latent variable* $\mathbf{z}_{t,i}^l$ *at the* $l-$*th layer,* $\mathbf{z}_{t,i}^l$ *is component-wise identifiability, i.e., there exists* $\hat{\mathbf{z}}_{t,j}^l$ *and an invertible function* $h_t^l : \mathbb{R} \to \mathbb{R}$, *such that* $\hat{z}_{t,j}^l = h_t^l(z_{t,i}^l)$.

**Discussion.** Please refer to the proof in Appendix B.7. Compared with existing works [28, 29, 74, 42] that also leverage similar assumptions (i.e., conditional independence and sufficient variability assumptions) to achieve component-wise identification, the main difference is that our method simultaneously leverage the time-delayed and hierarchical parents to meet the sufficient variability condition instead of domain or historical information.

### 3.3 Comparison with Existing Results of Causal Representation Learning

Compared to recent methods for causal representation learning, the proposed CHiLD model better aligns with real-world scenarios from multiple perspectives, as summarized in Table 1. Specifically, TDRL [74] and PCL [29] struggle to handle time-series data with hierarchical latent dynamics and noisy mixing processes. Although CaRiNG [5] also leverages temporal contextual information for identification, it remains limited as it fails to account for noisy mixing processes and cannot identify latent dynamics with instantaneous dependencies. More recently, Zhang et al. [77] and Li et al. [48] have employed sparse causal influences to identify latent variables with instantaneous dependencies. However, these methods neither address the challenges posed by noisy mixing processes nor identify latent variables within hierarchical structures. Another closely related work, Fu et al. [14], also utilizes unique eigenvalue-eigenfunction decomposition to achieve identifiability. However, this method is specifically designed for causal discovery in climate data and does not account for hierarchical latent dynamics. In contrast, our model extends previous results by establishing the relationship between the minimal required temporal observations and the size of hierarchical latent variables.

## 4 Approach

In this section, we propose the VAE-based model [39, 22] for time series generation as shown in Figure 2. Specifically, we begin by estimating the hierarchical latent variables using a contextual encoder which processes a sequence of observations, $\mathbf{x}_{t-L:t+L}$ [5], and outputs the estimated latent variables $\hat{\mathbf{z}}_t$. Sequentially, the reconstructed observations are generated from the latent space via a step-wise decoder, which uses $\hat{\mathbf{z}}_t$ to produce $\hat{\mathbf{x}}_t$.

---

[5]The subscript denotes the observations from $t - L$ to $t + L$ time steps.

To capture the hierarchical latent dynamics, we employ the Kullback–Leibler divergence between the posterior distribution of the estimated latent variables and a prior distribution, which coincides with the natural sparsity of hierarchical structure in Theorem 2 and the conditional independence condition in Lemma 1. Moreover, we devise a hierarchical prior network to estimate the prior distribution of latent

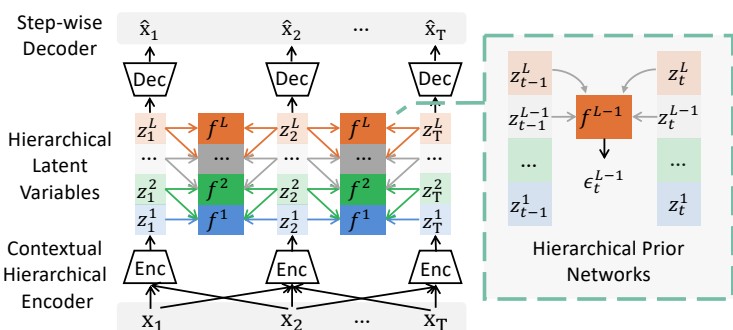

Figure 2: The overall framework of CHiLD, which incorporates contextual Hierarchical encoder (**Enc**), step-wise decoder (**Dec**), and hierarchical prior networks.

variables, which is based on the normalizing flow [61, 63] and converts the prior distribution into a Gaussian noise like [74, 5]. Please refer to Appendix F.3 for implementation details.

## 4.1 Contextual Hierarchical Encoder and Step-wise Decoder

Guided by Theorem 1, we use the temporal context observations to identify multi-layer latent variables. Therefore, we devise a contextual hierarchical encoder for the inference process in Equation (7):

$$\hat{\mathbf{z}}_t^1 = \phi_1(\mathbf{x}_{t-L:t+L}), \cdots, \hat{\mathbf{z}}_t^l = \phi_l(\hat{\mathbf{z}}_t^{l-1}), \cdots, \hat{\mathbf{z}}_t^L = \phi_L(\hat{\mathbf{z}}_t^{L-1}) \tag{7}$$

which $\phi_1$ encodes the observations to the first latent layer, and adopts a similar form of 1D convolutional neural networks. Subsequently, each $\phi_L$ encode the preceding latent layer, $\hat{\mathbf{z}}_t^{L-1}$, to extract the deeper or higher-level components, yielding $\hat{\mathbf{z}}_t^L$. $L$ can be considered as the size of the receptive field. Accordingly, we employ a neural architecture based on temporal convolutional neural networks (TCN) [44, 45] as the contextual encoder. Given the estimated latent variables $\hat{\mathbf{z}}_t$, we further develop a step-wise decoder that only takes the bottom-layer $\hat{\mathbf{z}}_t$ as input to generate $\hat{\mathbf{x}}_t$ for each time-step within the receptive field, as shown in Equation (8).

$$\hat{\mathbf{x}}_t = \psi(\hat{\mathbf{z}}_t). \tag{8}$$

## 4.2 Hierarchical Prior Networks

To make the estimated latent variables capture the hierarchical property and the conditional independent property , one straightforward solution is to minimize the Kullback–Leibler divergence between the posterior distribution and the prior distribution. However, the prior distribution is unknown and if we estimate the prior density directly, arbitrary density functions may be generated, leading to an incorrect posterior distribution. To address this challenge, motivated by [74, 5], who use normalizing flow [61] to model the causal procedure of latent variables by restricting the independence of estimated noise, we propose the hierarchical prior networks to estimate the prior distribution of latent variables. Specifically, given time delay $\tau$, we let $r_t^l = \{r_{t,1}^l, \cdots, r_{t,i}^l, \cdots, r_{t,n}^l\}$ be a set of learned inverse transition functions that take $\hat{z}_{t,i}^l$, $\hat{z}_{t-\tau}^l$, and $\hat{z}_t^{l+1}$ as input and generates the noise term, i.e., $\hat{\epsilon}_{t,i}^l = r_{t,i}^l(\hat{z}_{t,i}^l, \hat{z}_{t-\tau}^l, \hat{z}_t^{l+1})$. And $r_t^l$ is implemented by MLPs. Sequentially, we can devise a transformation $\kappa : \{\hat{\mathbf{z}}_{t-\tau}^l, \hat{\mathbf{z}}_t^{l+1}, \hat{\mathbf{z}}_t^l\} \to \{\hat{\mathbf{z}}_{t-\tau}^l, \hat{\mathbf{z}}_t^{l+1}, \hat{\epsilon}_t^l\}$ with its corresponding Jacobian

$$\mathbf{J}_\kappa = \begin{pmatrix} \mathbb{I} & 0 & 0 \\ 0 & \mathbb{I} & 0 \\ * & * & \mathrm{diag}\left(\frac{\partial r_{t,i}^l}{\partial \hat{z}_{t,i}^l}\right) \end{pmatrix}, \text{ where } * \text{ denotes a matrix. By applying the change of variables}$$

formula, we have:

$$p(\hat{\mathbf{z}}_{t-\tau}^l, \hat{\mathbf{z}}_t^{l+1}, \hat{\mathbf{z}}_t^l) = p(\hat{\mathbf{z}}_{t-\tau}^l, \hat{\mathbf{z}}_t^{l+1}, \hat{\epsilon}_t^l)|\mathbf{J}_\kappa|. \tag{9}$$

By dividing $p(\hat{\mathbf{z}}_{t-\tau}^l, \hat{\mathbf{z}}_t^{l+1})$ on both sides, the prior distribution can be derived as follows:

$$\log p(\hat{\mathbf{z}}_t^l|\hat{\mathbf{z}}_{t-\tau}^l, \hat{\mathbf{z}}_t^{l+1}) = \log p(\hat{\epsilon}_t^l|\hat{\mathbf{z}}_{t-\tau}^l, \hat{\mathbf{z}}_t^{l+1}) + \log|\mathbf{J}_\kappa| = \sum_i^n \left( \log p(\hat{\epsilon}_{t,i}^l) + \log \frac{\partial \hat{\epsilon}_{t,i}^l}{\partial \hat{z}_{t,i}^l} \right), \tag{10}$$

Table 2: Experimental results of MCC on simulation data

| Dataset | CHiLD | IDOL | CaRiNG | TDRL | Beta-VAE | SlowVAE | iVAE | FactorVAE | PCL | TCL |
|---------|-------|------|--------|------|----------|---------|------|-----------|-----|-----|
| A | **0.901(0.043)** | 0.842(0.033) | 0.836(0.045) | 0.823(0.025) | 0.750(0.005) | 0.774(0.022) | 0.604(0.023) | 0.509(0.031) | 0.513(0.027) | 0.405(0.021) |
| B | 0.941(0.037) | 0.938(0.004) | 0.941(0.001) | **0.942(0.076)** | 0.753(0.045) | 0.858(0.065) | 0.524(0.054) | 0.417(0.055) | 0.384(0.011) | 0.404(0.008) |
| C | **0.835(0.012)** | 0.801(0.031) | 0.802(0.029) | 0.797(0.018) | 0.748(0.017) | 0.742(0.087) | 0.615(0.023) | 0.322(0.098) | 0.329(0.034) | 0.356(0.012) |
| D | **0.841(0.016)** | 0.765(0.055) | 0.788(0.082) | 0.771(0.003) | 0.708(0.015) | 0.738(0.057) | 0.710(0.090) | 0.679(0.048) | 0.452(0.004) | 0.462(0.010) |
| E | **0.862(0.026)** | 0.722(0.057) | 0.728(0.023) | 0.702(0.032) | 0.716(0.024) | 0.515(0.077) | 0.723(0.040) | 0.741(0.043) | 0.621(0.027) | 0.069(0.058) |
| F | **0.808(0.002)** | 0.687(0.029) | 0.723(0.024) | 0.678(0.078) | 0.667(0.034) | 0.685(0.066) | 0.691(0.035) | 0.630(0.009) | 0.384(0.150) | 0.319(0.024) |
| G | **0.774(0.007)** | 0.609(0.075) | 0.572(0.044) | 0.505(0.096) | 0.523(0.017) | 0.429(0.009) | 0.611(0.005) | 0.499(0.003) | 0.442(0.021) | 0.345(0.039) |

where $p(\hat{\epsilon}_{t,i}^l)$ follow Gaussian distributions. For the reconstruction likelihood $\mathcal{L}_R$, we use the mean-squared error (MSE) to measure the discrepancy between the generated and original observations.

**Further Discussion:** Please note that although CaRiNG [5] also leverages the contextual observation in a VAE-based framework, the main difference between CaRiNG and CHiLD can be summarized into the following two folds. **1)** The model design comes from different theoretical guidance. Motivated by the nonlinear ICA-based theoretical results, CaRiNG leverages the contextual observations to recover the lost latent information and address the challenge of a non-invertible mixing function. In contrast, our method is guided by the uniqueness of spectral decomposition and overcomes the difficulties of hierarchical latent dynamics. **2)** The prior distribution estimations of these two methods are also different. CaRiNG assumes single-layer latent variables and uses the time-delayed and current latent variables to estimate noise. Meanwhile, the CHiLD allows hierarchical latent dynamics and hence harnesses the current, time-delayed, as well as hierarchical latent variables for noise estimation.

## 4.3  Optimization

Finally, we train the proposed model by optimizing the Evidence Lower Bound (ELBO) as follows

$$
ELBO = \mathbb{E}_{q(\mathbf{z}_{1:T}|\mathbf{x}_{1:T})} \ln p(\mathbf{x}_{1:T}|\mathbf{z}_{1:T}) - D_{KL}(q(\mathbf{z}_{1:T}|\mathbf{x}_{1:T})||p(\mathbf{z}_{1:T}))
$$

$$
= \underbrace{\mathbb{E}_{q(\mathbf{z}_{1:T}|\mathbf{x}_{1:T})} \sum_{t=1}^{T} \log p(\mathbf{x}_t|\mathbf{z}_t)}_{\mathcal{L}_R} + \underbrace{\mathbb{E}_{q(\mathbf{z}_{1:T}|\mathbf{x}_{1:T})} \left[ \sum_{t=1}^{T} \Big( \log p(\mathbf{z}_t|\mathbf{z}_{t-1:t-\tau}) - \log q(\mathbf{z}_t|\mathbf{x}_{t-L:t+L}) \Big) \right]}_{\mathcal{L}_{KL}},
$$

(11)

where $D_{KL}$ denotes the Kullback–Leibler divergence and the detailed derivation can be found in Appendix F.2. For the reconstruction loss $\mathcal{L}_R$, we use the mean-squared error (MSE) to measure the discrepancy between the generated and original observations. And we leverage the Equation (10) to approximate the KL divergence.

## 5  Experiment

### 5.1  Experiments on Synthetic Data

**Data Generation.** We generate synthetic time series data based on a fixed latent causal process, as defined in Equations (1)–(3). To evaluate the proposed theoretical results, we provide seven different datasets from A to G, with different numbers of latent variables, different complexities of generation processes, and different latent layers. Note that Dataset B consists of single-layer latent variables, while other datasets adhere to the assumptions underlying the proposed theoretical framework. Details of the data generation process and evaluation metrics can be found in Appendix G.1.

**Baselines.** To evaluate the effectiveness of our method, we consider the following baselines. First, we consider the standard $\beta$-VAE [20] and FactorVAE [38], which do not use historical information. Then we consider TDRL [74], iVAE [37], TCL [28], PCL [29], and SlowVAE [40], which use temporal information but do not contain instantaneous dependency. We also consider CaRiNG [5] and IDOL [48], which leverage contextual observations and consider instantaneous dependency, respectively. We use mean correlation coefficient (MCC) as the evaluation metric.

**Results and Discussion.** Experiment results are shown in Table 2. According to the experiment results, we can find that: 1) the proposed CHiLD achieves the highest MCC performance in Datasets A, C, D, and E; 2) the CHiLD and the recently proposed methods like IDOL and CaRiNG achieve good performance in Dataset B since it does not contain hierarchical latent dynamics. 3) In the more challenging Dataset F and G (datasets with large dimensions and latent layers), although the

Table 3: Experiment results of real-world datasets for the time series generation.

| | Weather | WeatherBench | CESM2 | Box | Gestures | Throwcatch | Discussion | Purchases | WalkDog |
|---|---|---|---|---|---|---|---|---|---|
| | | | | Context-FID | | | | | |
| CHiLD | **0.507(0.042)** | **0.078(0.017)** | **0.018(0.003)** | 0.092(0.011) | **0.032(0.004)** | **0.022(0.003)** | **0.029(0.005)** | **0.021(0.009)** | **0.016(0.005)** |
| KoVAE | 2.131(0.637) | 0.171(0.032) | 0.198(0.030) | **0.056(0.009)** | 0.079(0.023) | 0.183(0.068) | 0.235(0.036) | 0.143(0.052) | 0.137(0.034) |
| Diffusion-TS | 3.826(0.915) | 1.712(0.163) | 0.407(0.037) | 0.258(0.017) | 0.088(0.006) | 0.151(0.011) | 0.487(0.116) | 0.300(0.048) | 0.334(0.037) |
| TimeGAN | 6.434(1.398) | 1.564(0.269) | 3.255(0.232) | 0.959(0.087) | 0.830(0.081) | 3.810(0.345) | 3.761(0.393) | 1.829(0.225) | 3.838(0.440) |
| IDOL | 1.694(0.485) | 0.239(0.030) | 0.142(0.017) | 0.125(0.026) | 0.018(0.004) | 0.025(0.002) | 0.082(0.033) | 0.029(0.014) | 0.051(0.022) |
| cwVAE | 5.061(1.359) | 0.409(0.056) | 0.238(0.044) | 0.215(0.026) | 0.329(0.051) | 0.284(0.025) | 0.919(0.132) | 1.147(0.146) | 0.565(0.126) |
| TimeVAE | 3.910(0.858) | 0.221(0.029) | 0.108(0.029) | 0.203(0.032) | 0.438(0.163) | 0.414(0.134) | 0.067(0.020) | 0.074(0.008) | 0.082(0.012) |
| | | | | Correlational Score | | | | | |
| CHiLD | **0.165(0.004)** | **5.190(0.016)** | **1.250(0.009)** | **1.599(0.021)** | **0.089(0.003)** | **0.065(0.001)** | **0.014(0.000)** | **0.100(0.003)** | **0.126(0.001)** |
| KoVAE | 1.620(0.012) | 20.384(0.082) | 10.087(0.017) | 1.758(0.038) | 1.015(0.030) | 1.268(0.051) | 4.060(0.023) | 2.689(0.054) | 3.015(0.021) |
| Diffusion-TS | 1.932(0.021) | 63.291(0.630) | 16.659(0.231) | 3.407(0.125) | 1.724(0.220) | 2.101(0.243) | 5.851(0.109) | 3.513(0.157) | 3.701(0.037) |
| TimeGAN | 3.438(0.020) | 57.475(1.213) | 20.641(0.082) | 7.373(0.937) | 10.917(1.620) | 16.419(1.334) | 27.876(0.485) | 25.362(1.868) | 21.848(0.230) |
| IDOL | 1.195(0.014) | 34.773(0.059) | 9.125(0.016) | 1.849(0.035) | 0.634(0.008) | 0.559(0.007) | 1.295(0.010) | 1.148(0.008) | 2.067(0.016) |
| cwVAE | 0.584(0.013) | 34.690(0.131) | 6.469(0.022) | 0.812(0.026) | 0.926(0.064) | 0.755(0.025) | 8.624(0.032) | 7.741(0.055) | 6.367(0.079) |
| TimeVAE | 0.707(0.005) | 24.873(0.091) | 6.363(0.022) | 0.891(0.018) | 1.046(0.047) | 1.063(0.046) | 0.810(0.002) | 0.568(0.002) | 0.664(0.003) |

(a) CHiLD          (b) TimeVAE          (c) DiffusionTS          (d)TimeGAN

Figure 3: Interpolation visualization of different models. For each method, after training the model, only one latent variable is gradually changed while keeping the other variables fixed. The images of each method from left to right represent the gradual increase of the latent variable.

MCC performance of the proposed CHiLD is lower, it can still outperform the existing methods for temporal causal representation learning, like IDOL, which supports our theoretical claims.

## 5.2 Experiments on Real-world Data

### 5.2.1 Experiment Setup

**Datasets.** We consider the task of unconditional time series generation and use the following datasets: Stock, ETTh1, fMRI, MuJoCo, and two human motion datasets, Human3.6M and HumanEva-I. For Human3.6M [32], we choose 3 motions: Discussion, Purchases, and WalkingDog. For HumanEva-I [65], we choose Box, Gesture, and Throwcatch. We further consider three climate datasets: Weather, WeatherBench, and CESM2. Please refer to Appendix G.2.1 and D for the dataset description and the connection between time series generation and modeling hierarchical temporal latent dynamics. **Baselines.** First, we consider the VAE-based methods like cwVAE [64], KoVAE [58] and TimeVAE [9]. We also consider other types of generative models like Diffusion-TS [76] and TimeGAN [75]. Moreover, we consider the causal representation-based method like IDOL [48]. We choose Context-Fréchet Inception Distance (Context-FID) and the Correlational Score as evaluation metrics. We repeat each experiment over 3 random seeds and publish the mean and standard deviation. Please refer to Appendix G.2.2 for the introduction to the evaluation metric.

### 5.2.2 Quantitative Results

The experimental results in Table 3 show that the proposed CHiLD method significantly outperforms all baselines across most datasets, such as Box and Discussion, highlighting its potential for time series generation. While TimeVAE and KoVAE also employ VAE-based architectures, they assume independent latent variables. In contrast, CHiLD enforces a hierarchical latent structure, enabling it to model high-level dependence and hence generate more realistic time series data. Other time series generation baselines, such as Diffusion-TS and TimeGAN, fail to deliver optimal performance, particularly on human motion datasets. This can be attributed to the higher dimensionality of human motion data compared to datasets like ETTh1, which implies more complex latent dynamics. Please refer to Appendix G.3.2 and G.3.1 for more experiment results and the ablation studies.

### 5.2.3 Qualitative Results

We also provide the interpolation visualization results. Specifically, after training each model, we change one latent variable gradually and keep the other fixed.

**Controllable Generation.** To evaluate controllable generation, we compare TimeVAE, TimeGAN, and DiffusionTS in the Discussion motion generation dataset. As shown in Figure 3. TimeGAN performs the worst, as the generated motion patterns are difficult to recognize as human shapes. TimeVAE produces more structured movements but lacks coordination. While DiffusionTS generates recognizable human shapes, the variations in motion with respect to the latent variable are limited, indicating weak controllability. In contrast, CHiLD method generates high-quality human motions and enables coordinated changes in limb movements through latent variable control. Please find more interpolation results in the form of GIF visualization from the supplementary material.

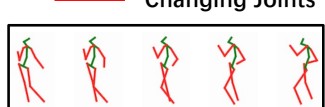 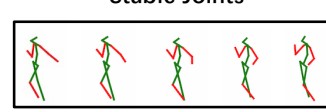

(a) Interpolation visualizations of High-Level Latent Variables

(b) Interpolation Visualizations of Low-Level Latent Variables

Figure 4: Interpolation visualization of high-level and low-level latent variables of the proposed CHiLD method. Interpolating high-level latent variables leads to more significant changes in observed data compared to interpolating low-level latent variables.

**Hierarchical Generation.** We interpolate both high-level and low-level latent variables in CHiLD and generate corresponding motions. As shown in Figure 4, green represents stable joints, while red indicates joints that change with the latent variable. The results show that high-level latent variables influence a broader range of joints compared to low-level ones, demonstrating that our method effectively captures hierarchical abstractions in time-series data.

# 6 Conclusion

This paper introduces an identification framework for temporal causal representation learning under hierarchical latent dynamics. The proposed method leverages temporal contextual observations to estimate the joint distributions of multi-layer latent variables and exploits the sparsity of hierarchical structures to achieve precise latent variable identification within each layer. A VAE-based generative model, incorporating a contextual encoder and normalizing flow-based hierarchical priors, enforces independent noise conditions while enhancing the capacity to generate realistic time series data. Empirical evaluations on both synthetic and real-world datasets validate the theoretical guarantees and demonstrate the effectiveness of our approach. Future works will extend this framework to more complex domains, including controllable video generation, as well as applications in neuroscience, finance, and climate modeling. **Limitation:** However, the component-wise identifiability result of our work also relies on other assumptions like layer-wise conditional independence. How to further relax these assumptions would be an interesting future direction.

# 7 Acknowledgment

The authors would like to thank the anonymous reviewers for helpful comments and suggestions during the reviewing process. The authors would like to acknowledge the support from NSF Award No. 2229881, AI Institute for Societal Decision Making (AI-SDM), the National Institutes of Health (NIH) under Contract R01HL159805, and grants from Quris AI, Florin Court Capital, and MBZUAI-WIS Joint Program, and the Al Deira Causal Education project. Moreover, this research was supported in part by National Science and Technology Major Project (2021ZD0111501), National Science Fund for Excellent Young Scholars (62122022), Natural Science Foundation of China (U24A20233, 62206064, 62206061, 62476163, 62406078), Guangdong Basic and Applied Basic Research Foundation (2024A1515011901, 2023A04J1700, 2023B1515120020).

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

# A    Useful Definitions and Lemmas

**Definition 4.** *(Diagonal Operator) Consider two random variable $a$ and $b$, density functions $p_a$ and $p_b$ are defined on some support $\mathcal{A}$ and $\mathcal{B}$, respectively. The diagonal operator $D_{b|a}$ maps the density function $p_a$ to another density function $D_{b|a} \circ p_a$ defined by the pointwise multiplication of the function $p_{b|a}$ at a fixed point $b$:*

$$p_{b|a}(b \mid \cdot)p_a = D_{b|a} \circ p_a, where\ D_{b|a} = p_{b|a}(b \mid \cdot). \tag{A1}$$

**Definition 5.** *(Completeness) A family of distribution $p(a|b)$ is complete if the only solution $p(a)$ to*

$$\int_A p(a)p_{a|b}(a|b)\,da = 0 \quad for\ all\ b \in \mathcal{B}, \tag{A2}$$

is $p(a) = 0$. In other words, no matter the range of an operator is on finite or infinite, it is complete if its null space[6] or kernel is a zero set. Completeness is always used to phrase the sufficient and necessary condition for injective linear operator [59, 6].

**Theorem A1.** *(Theorem XV 4.5 in [11] Part III) A bounded operator $T$ is a spectral operator if and only if it is the sum $T = S + N$ of a bounded scalar type operator $S$ and a quasi-nilpotent operator $N$ commuting with $S$. Furthermore, this decomposition is unique and $T$ and $S$ have the same spectrum and the same resolution of the identity.*

**Lemma A1.** *(Lemma 1 in [24]) Under Assumption 1, if $L_{z|x}$ is injective, then $L_{x|z}^{-1}$ exists and is densely defined over $\mathcal{G}(\mathcal{X})$ (for $\mathcal{G} = \mathcal{L}^1, \mathcal{L}_{\mathrm{bnd}}^1$).*

**Properties of linear operator.**    We outline useful properties of the linear operator to facilitate understanding of our proof:

i. *(Inverse)* If linear operator: $L_{b|a}$ exists a left-inverse $L_{b|a}^{-1}$, such $L_{b|a}^{-1} \circ L_{b|a} \circ p_a = p_a$ for all $a \in \mathcal{A}$. Analogously, if $L_{b|a}$ exists a right-inverse $L_{b|a}^{-1}$, such $L_{b|a} \circ L_{b|a}^{-1} \circ p_a = p_a$ for all $a \in \mathcal{A}$. If $L_{b|a}$ is bijective, there exists left-inverse and right-inverse which are the same.

ii. *(Injective)* $L_{b|a}$ is said to be an injective linear operator if its $L_{b|a}^{-1}$ is defined over the range of the operator $L_{b|a}$ [43]. If so, under assumption 1 (ii ), $L_{a|b}^{-1}$ exists and is densely defined over $\mathcal{F}(\mathcal{A})$. [24].

iii. *(Composition)* Given two linear operators $L_{c|b} : \mathcal{F}(\mathcal{B}) \to \mathcal{F}(\mathcal{C})$ and $L_{c|a} : \mathcal{F}(\mathcal{A}) \to \mathcal{F}(\mathcal{C})$, with the function space supports defined uniformly on the range of supports for the domain spaces as characterized by $L_{b|a}$, it follows that $L_{c|a} = L_{c|b} \circ L_{b|a}$. Furthermore, the properties of linearity and associativity are preserved in the operation of linear operators. However, it is crucial to note the non-commutativity of these operators, i.e., $L_{c|b}L_{b|a} \neq L_{b|a}L_{c|b}$, indicating the significance of the order of application.

**Definition 6** (Markov Network). *Markov network is an undirected graph $G = (V, E)$ with a set of random variables $\mathbf{x}_{v \in V}$, where any two non-adjacent variables like $x_a$ and $x_b$ are conditionally independent given all other variables. That is,*

$$x_a \perp x_b | \mathbf{x}_{V \setminus \{a,b\}}, \quad \forall (a, b) \notin E. \tag{A3}$$

Markov Networks and Directed Acyclic Graphs (DAGs) are both graphical models employed to represent joint distributions and to illustrate conditional independence properties. Based on the definition of Markov Network, we further define Vector-Node Markov Network as follows:

**Definition 7** (Vector-Node Markov Network). *A Markov network where each node represents a vector, rather than a scalar variable. In this structure, the joint distribution is factorized over a graph $G = (V, E)$, where $V$ is the set of nodes, and each $v \in V$ corresponds to a multidimensional vector $\mathbf{x}_v$. Given any two non-adjacent vectors $\mathbf{x}_a$ and $\mathbf{x}_b$, any $\mathbf{x}_{a,i} \in \mathbf{x}_a$ and $\mathbf{x}_{b,j} \in \mathbf{x}_b$ are conditionally independent given all other variables. That is,*

$$\mathbf{x}_{a,i} \perp \mathbf{x}_{b,j} | \mathbf{x}_{V \setminus \{a,b\}}, \quad \forall (a, b) \notin E. \tag{A4}$$

---

[6]The null space or kernel of an operator $L$ to be the set of all vectors which $L$ maps to the zero vector: null $L = \{v \in V : Lv = 0\}$.

**Definition 8** (Isomorphism of Markov networks). *We let the $V(\cdot)$ be the vertical set of any graphs, an isomorphism of Markov networks $M$ and $\hat{M}$ is a bijection between the vertex sets of $M$ and $\hat{M}$*

$$f : V(M) \to V(\hat{M})$$

*such that any two vertices $u$ and $v$ of $M$ are adjacent in $G$ if and only if $f(u)$ and $f(v)$ are adjacent in $\hat{M}$.*

**Definition 9** (**Definition of left and right inverse.**). *Let $f : \mathbf{x} \to \mathbf{y}$ and $g : \mathbf{y} \to \mathbf{x}$ be functions. We say that $g$ is a left inverse to $f$, and that $f$ is a right inverse of $g$, if $g \circ f = id_X$, where $id_X$ is the identity function.*

# B   Proof of Theoretical Results

## B.1   Notations

This section collects the notations used in the theorem proofs for clarity and consistency.

Table A4: List of notations, explanations, and corresponding values.

| Index | Explanation | Support |
|---|---|---|
| $n$ | Number of variables | $n \in \mathbb{N}^+$ |
| $i, j, k, s, o$ | Index of latent variables | $i, j, k, s, o \in \{1, \cdots, n\}$ |
| $t$ | Time index | $t \in \mathbb{N}^+$ |
| $L$ | Number of latent layers | $L \in \mathbb{N}^+$ and $L \geq 2$ |
| $l$ | Index of latent layers | $l = \{1, 2, \cdots, L\}$ |
| **Variable** | | |
| $\mathcal{X}_t$ | Support of observed variables in time-index $t$ | $\mathcal{X}_t \subseteq \mathbb{R}^n$ |
| $\mathcal{Z}_t$ | Support of latent variables | $\mathcal{Z}_t \subseteq \mathbb{R}^n$ |
| $\mathbf{x}_t$ | Observed variables in time-index $t$ | $\mathbf{x}_t \in \mathbb{R}^n$ |
| $\mathbf{z}_t$ | Latent variables in time-index $t$ | $\mathbf{z}_t \in \mathbb{R}^n$ |
| $z_{t,i}^l$ | The $i$-th, $l$-th-layer latent variable at $t$-th step | $z_{t,i}^l \in \mathbb{R}$ |
| $\mathbf{c}_t$ | $\{\mathbf{z}_{t-1}, \mathbf{z}_t\}$ | $\mathbf{c}_t \in \mathbb{R}^{2 \times n \times L}$ |
| $\boldsymbol{\epsilon}_t^0$ | Independent noise of mixing procedure | $\boldsymbol{\epsilon}_t^0 \sim p_{\epsilon_t^0}$ |
| $\boldsymbol{\epsilon}_{t,i}^l$ | Independent noise of the latent transition of $\mathbf{z}_{t,i}^l$ | $\boldsymbol{\epsilon}_{t,i}^l \sim p_{\epsilon_{t,i}^l}$ |
| **Function** | | |
| $p_{a\mid b}(\cdot \mid b)$ | Density function of $a$ given $b$ | / |
| $p_{a,b\mid c}(a, \cdot \mid c)$ | Joint density function of $(a, b)$ given $a$ and $c$ | / |
| $\mathbf{pa}_d(\cdot)$ | Time-delayed parents | / |
| $\mathbf{pa}_h(\cdot)$ | Hierarchical parents | / |
| $\mathbf{g}(\cdot)$ | Nonlinear mixing function | $\mathbb{R}^n \to \mathbb{R}^n$ |
| $f_i^l(\cdot)$ | Transition function of $\mathbf{z}_{t,i}^l$ | $\mathbb{R}^{\lvert \mathrm{Pa}_d(\mathbf{z}_{t,i}^l)\rvert + \lvert \mathrm{Pa}_h(\mathbf{z}_{t,i}^l)\rvert + 1} \to \mathbb{R}$ |
| $h(\cdot)$ | Invertible transformation from $\mathbf{z}_t$ to $\hat{\mathbf{z}}_t$ | $\mathbb{R}^n \to \mathbb{R}^n$ |
| $\pi(\cdot)$ | Permutation function | $\mathbb{R}^n \to \mathbb{R}^n$ |
| $\mathcal{F}$ | Function space | / |
| $\mathcal{M}_{\mathbf{c}_t}$ | Markov network over $\mathbf{c}_t$ | / |
| $\phi_l$ | Contextual hierarchical encoder for $\hat{\mathbf{z}}_t^l$ | / |
| $\psi_l$ | Step-wise Decoder | / |
| $r_{t,i}^l$ | Noise estimator of $\hat{\epsilon}_{t,i}^l$. | / |
| **Symbol** | | |
| $\mathbf{J}_\kappa$ | Jacobian matrix of $r_t^l$ | / |

## B.2   Proof of Block-wise Identifiability of Latent Variables $\mathbf{z}_t$ in Hierarchical Latent Process

**Theorem A2.** (*Block-wise Identifiability of Latent Variables $\mathbf{z}_t$ in Hierarchical Latent Process.*)
*Suppose the observed and $L$-layer latent variables follow the data generation process in Figure*

*1. By matching the true joint distribution of $2L + 1$ number of adjacent observed variables, i.e., $\{\mathbf{x}_{t-L}, \cdots, \mathbf{x}_t, \cdots, \mathbf{x}_{t+L}\}$, we further make the following assumptions:*

- *(i) The joint distribution of $X, Z$, and their marginal and conditional densities are bounded and continuous.*

- *(ii) The linear operators $L_{\mathbf{x}_{t+L}, \cdots, \mathbf{x}_{t+1} | \mathbf{z}_t}$ and $L_{\mathbf{x}_{t-1}, \cdots, \mathbf{x}_{t-L} | \mathbf{x}_{t+1}, \cdots, \mathbf{x}_{t+L}}$ are injective for bounded function space.*

- *(iii) For all $\mathbf{z}_t, \mathbf{z}_t' \in \mathcal{Z}_t (\mathbf{z}_t \neq \mathbf{z}_t')$, the set $\{\mathbf{x}_t : P(\mathbf{x}_t | \mathbf{z}_t) \neq P(\mathbf{x}_t | \mathbf{z}_t')\}$ has positive probability.*

*Suppose that we have learned $(\hat{g}, \hat{f}, P(\hat{\mathbf{z}}_t))$ to achieve Equation (1) and (2), then the latent variables $\mathbf{z}_t = \{\mathbf{z}_t^1, \cdots, \mathbf{z}_t^L\}$ are block-wise identifiable.*

*Proof.* We first follow Hu et al. [24] framework to prove that $\mathbf{z}_t$ is block-wise identifiable given sufficient observation. Sequentially, we prove that we require at least $2L + 1$ adjacent observed variables to achieve block-wise identifiability.

Given time series data with $T$ timesteps $X = \{\mathbf{x}_1, \cdots, \mathbf{x}_t, \cdots, \mathbf{x}_T\}$ and $L$-layer latent variables, we let $\mathbf{x}_{<t}$ and $\mathbf{x}_{>t}$ be $\{\mathbf{x}_1, \cdots, \mathbf{x}_{t-1}\}$ and $\{\mathbf{x}_{t+1}, \cdots, \mathbf{x}_T\}$, respectively. Note that the length of $\mathbf{x}_{<t}$ and $\mathbf{x}_{>t}$ are larger than $L$, i.e., $|\mathbf{x}_{<t}| > L$ and $|\mathbf{x}_{>t}| > L$. Sequentially, according to the data generation process in Figure 1, we have:

$$P(\mathbf{x}_{<t} | \mathbf{x}_t, \mathbf{z}_t) = P(\mathbf{x}_{<t} | \mathbf{z}_t), \quad P(\mathbf{x}_{>t} | \mathbf{x}_t, \mathbf{x}_{<t}, \mathbf{z}_t) = P(\mathbf{x}_{>t} | \mathbf{z}_t). \tag{A5}$$

Sequentially, the observed $P(\mathbf{x}_{t-1})$ and joint distribution $P(\mathbf{x}_{>t}, \mathbf{x}_t, \mathbf{x}_{<t})$ directly indicates $P(\mathbf{x}_{>t}, \mathbf{x}_t | \mathbf{x}_{<t})$, and we have:

$$P(\mathbf{x}_{>t}, \mathbf{x}_t | \mathbf{x}_{<t}) = \underbrace{\int_{\mathcal{Z}_t} P(\mathbf{x}_{>t}, \mathbf{x}_t, \mathbf{z}_t | \mathbf{x}_{<t}) d\mathbf{z}_t}_{\text{Integration over } \mathcal{Z}_t} = \underbrace{\int_{\mathcal{Z}_t} P(\mathbf{x}_{>t} | \mathbf{x}_t, \mathbf{z}_t, \mathbf{x}_{<t}) P(\mathbf{x}_t, \mathbf{z}_t | \mathbf{x}_{<t}) d\mathbf{z}_t}_{\text{Factorization of joint conditional probability}}$$

$$= \underbrace{\int_{\mathcal{Z}_t} P(\mathbf{x}_{>t} | \mathbf{z}_t) P(\mathbf{x}_t, \mathbf{z}_t | \mathbf{x}_{<t}) d\mathbf{z}_t}_{\text{Conditional Independence}} = \underbrace{\int_{\mathcal{Z}_t} P(\mathbf{x}_{>t} | \mathbf{z}_t) P(\mathbf{x}_t | \mathbf{z}_t, \mathbf{x}_{<t}) P(\mathbf{z}_t | \mathbf{x}_{<t}) d\mathbf{z}_t}_{\text{Bayes Law}}$$

$$= \int_{\mathcal{Z}_t} P(\mathbf{x}_{>t} | \mathbf{z}_t) P(\mathbf{x}_t | \mathbf{z}_t) P(\mathbf{z}_t | \mathbf{x}_{<t}) d\mathbf{z}_t. \tag{A6}$$

We further incorporate the integration over $\mathcal{X}_{<t}$ as follows:

$$\int_{\mathcal{X}_{<t}} P(\mathbf{x}_{>t}, \mathbf{x}_t | \mathbf{x}_{<t}) P(\mathbf{x}_{<t}) d\mathbf{x}_{<t} = \int_{\mathcal{X}_{<t}} \int_{\mathcal{Z}_t} P(\mathbf{x}_{>t} | \mathbf{z}_t) P(\mathbf{x}_t | \mathbf{z}_t) P(\mathbf{z}_t | \mathbf{x}_{<t}) P(\mathbf{x}_{<t}) d\mathbf{z}_t d\mathbf{x}_{<t}. \tag{A7}$$

According to the definition of linear operator, we have:

$$\int_{\mathcal{X}_{<t}} P(\mathbf{x}_{>t}, \mathbf{x}_t | \mathbf{x}_{<t}) P(\mathbf{x}_{<t}) d\mathbf{x}_{<t} = [L_{\mathbf{x}_{>t}, \mathbf{x}_t | \mathbf{x}_{<t}} \circ P](\mathbf{x}_{<t}),$$

$$\int_{\mathcal{X}_{<t}} P(\mathbf{z}_t | \mathbf{x}_{<t}) P(\mathbf{x}_{<t}) d\mathbf{x}_{<t} = [L_{\mathbf{z}_t | \mathbf{x}_{<t}} \circ P](\mathbf{x}_{<t}) \tag{A8}$$

$$\int_{\mathcal{Z}_t} P(\mathbf{x}_{>t} | \mathbf{z}_t) d\mathbf{z}_t = L_{\mathbf{x}_{>t} | \mathbf{z}_t}.$$

By combining Equation (A7) and (A8), we have:

$$[L_{\mathbf{x}_{>t}, \mathbf{x}_t | \mathbf{x}_{<t}} \circ P](\mathbf{x}_{<t}) = [L_{\mathbf{x}_{>t} | \mathbf{z}_t} D_{\mathbf{x}_t | \mathbf{z}_t} L_{\mathbf{z}_t | \mathbf{x}_{<t}} \circ P](\mathbf{x}_{<t}), \tag{A9}$$

which implies the operator equivalence:

$$L_{\mathbf{x}_{>t}, \mathbf{x}_t | \mathbf{x}_{<t}} = L_{\mathbf{x}_{>t} | \mathbf{z}_t} D_{\mathbf{x}_t | \mathbf{z}_t} L_{\mathbf{z}_t | \mathbf{x}_{<t}}. \tag{A10}$$

Sequentially, we further integrate out $\mathbf{x}_t$ and have:

$$\int_{\mathcal{X}_t} L_{\mathbf{x}_{>t},\mathbf{x}_t|\mathbf{x}_{<t}} d\mathbf{x}_t = \int_{\mathcal{X}_t} L_{\mathbf{x}_{>t}|\mathbf{z}_t} D_{\mathbf{x}_t|\mathbf{z}_t} L_{\mathbf{z}_t|\mathbf{x}_{<t}} d\mathbf{x}_t, \tag{A11}$$

and it results in:

$$L_{\mathbf{x}_{>t}|\mathbf{x}_{<t}} = L_{\mathbf{x}_{>t}|\mathbf{z}_t} L_{\mathbf{z}_t|\mathbf{x}_{<t}}. \tag{A12}$$

According to assumption (ii), the linear operator $L_{\mathbf{x}_{>t}|\mathbf{z}_t}$ is injective, Equation (A12) can be rewritten as:

$$L_{\mathbf{x}_{>t}|\mathbf{z}_t}^{-1} L_{\mathbf{x}_{>t}|\mathbf{x}_{<t}} = L_{\mathbf{z}_t|\mathbf{x}_{<t}}. \tag{A13}$$

By combining Equation (A10) and (A13), we have

$$L_{\mathbf{x}_{>t},\mathbf{x}_t|\mathbf{x}_{<t}} = L_{\mathbf{x}_{>t}|\mathbf{z}_t} D_{\mathbf{x}_t|\mathbf{z}_t} L_{\mathbf{x}_{>t}|\mathbf{z}_t}^{-1} L_{\mathbf{x}_{>t}|\mathbf{x}_{<t}}. \tag{A14}$$

By leveraging Lemma A1, if $L_{\mathbf{x}_{<t}|\mathbf{x}_{>t}}$ is injective, then $L_{\mathbf{x}_{>t}|\mathbf{x}_{<t}}^{-1}$ exists. Therefore, we have:

$$L_{\mathbf{x}_{>t},\mathbf{x}_t|\mathbf{x}_{<t}} L_{\mathbf{x}_{>t}|\mathbf{x}_{<t}}^{-1} = L_{\mathbf{x}_{>t}|\mathbf{z}_t} D_{\mathbf{x}_t|\mathbf{z}_t} L_{\mathbf{x}_{>t}|\mathbf{z}_t}^{-1}. \tag{A15}$$

Then we can leverage assumption (iii) and the linear operator is bounded. Consequently, $L_{\mathbf{x}_{>t},\mathbf{x}_t|\mathbf{x}_{<t}} L_{\mathbf{x}_{>t}|\mathbf{x}_{<t}}^{-1}$ is also bounded, which satisfies the condition of Theorem A1, and hence the the operator $L_{\mathbf{x}_{>t}|\mathbf{z}_t} D_{\mathbf{x}_t|\mathbf{z}_t} L_{\mathbf{x}_{>t}|\mathbf{z}_t}^{-1}$ have a unique spectral decomposition, where $L_{\mathbf{x}_{>t}|\mathbf{z}_t}$ and $D_{\mathbf{x}_t|\mathbf{z}_t}$ correspond to eigenfunctions and eigenvalues, respectively.

Since both the marginal and conditional distributions of the observed variables are matched, the true model and the estimated model yield the same distribution over the observed variables. Therefore, we also have:

$$L_{\mathbf{x}_{>t},\mathbf{x}_t|\mathbf{x}_{<t}} L_{\mathbf{x}_{>t}|\mathbf{x}_{<t}}^{-1} = L_{\hat{\mathbf{x}}_{>t},\hat{\mathbf{x}}_t|\hat{\mathbf{x}}_{<t}} L_{\hat{\mathbf{x}}_{>t}|\hat{\mathbf{x}}_{<t}}^{-1}, \tag{A16}$$

where the L.H.S corresponds to the true model and the R.H.S corresponds to the estimated model. Moreover, $L_{\hat{\mathbf{x}}_{>t},\hat{\mathbf{x}}_t|\hat{\mathbf{x}}_{<t}} L_{\hat{\mathbf{x}}_{>t}|\hat{\mathbf{x}}_{<t}}^{-1}$ also have the unique decomposition, so the L.H.S of the Equation (A16) can be written as:

$$L_{\mathbf{x}_{>t},\mathbf{x}_t|\mathbf{x}_{<t}} L_{\mathbf{x}_{>t}|\mathbf{x}_{<t}}^{-1} = L_{\hat{\mathbf{x}}_{>t}|\hat{\mathbf{z}}_t} D_{\hat{\mathbf{x}}_t|\hat{\mathbf{z}}_t} L_{\hat{\mathbf{x}}_{>t}|\hat{\mathbf{z}}_t}^{-1}, \tag{A17}$$

Integrating Equation A15 and Equation A17, and noting that their L.H.S. are identical, it follows that they share the same spectral decomposition. This yields

$$L_{\mathbf{x}_{>t}|\mathbf{z}_t} = C L_{\hat{\mathbf{x}}_{>t}|\hat{\mathbf{z}}_t} P, \quad D_{\mathbf{x}_t|\mathbf{z}_t} = P^{-1} D_{\hat{\mathbf{x}}_t|\hat{\mathbf{z}}_t} P, \tag{A18}$$

where $C$ is a scalar accounting for scaling indeterminacy and $P$ is a permutation on the order of elements in $D_{\hat{\mathbf{x}}_t|\hat{\mathbf{z}}_t}$, as discussed in [11]. These forms of indeterminacy are analogous to those in eigendecomposition, which can be viewed as a finite-dimensional special case. P is a mapping from distribution to distribution

Since the normalizing condition

$$\int_{\hat{\mathcal{X}}_{t+1}} p_{\hat{\mathbf{x}}_t|\hat{\mathbf{z}}_t} d\hat{\mathbf{x}}_t = 1 \tag{A19}$$

must hold for every $\hat{\mathbf{z}}_t$, one only solution is to set $C = 1$.

Hence, $D_{\hat{\mathbf{x}}_t|\hat{\mathbf{z}}_t}$ and $D_{\mathbf{x}_t|\mathbf{z}_t}$ are identical up to a permutation on their repsective elements. We use unordered sets to express this equivalence:

$$\{p(\mathbf{x}_t \mid \mathbf{z}_t)\} = \{p(\mathbf{x}_t \mid \hat{\mathbf{z}}_t)\}, \quad \text{for all } \mathbf{z}_t, \hat{\mathbf{z}}_t. \tag{A20}$$

Due to the set being unordered, the only way to match the R.H.S. with the L.H.S. in a consistent order is to exchange the conditioning variables, that is,

$$\left\{ p(\mathbf{x}_t|\mathbf{z}_t^{(1)}), p(\mathbf{x}_t|\mathbf{z}_t^{(2)}), \cdots \right\} = \left\{ p(\mathbf{x}_t|\hat{\mathbf{z}}_t^{(\pi(1))}), p(\mathbf{x}_t|\hat{\mathbf{z}}_t^{(\pi(2))}), \cdots \right\}, \tag{A21}$$

where superscript $(\cdot)$ denotes the index of a conditioning variable, and $\pi$ is reindexing the conditioning variables. We use a relabeling map $h$ to represent its corresponding value mapping:

$$p(\mathbf{x}_t|\mathbf{z}_t) = p(\mathbf{x}_t|h(\hat{\mathbf{z}}_t)), \text{ for all } \mathbf{z}_t, \hat{\mathbf{z}}_t \tag{A22}$$

Since $K_{\hat{\mathbf{z}}_t, \mathbf{z}_t}$, $L_{\mathbf{x}_{>t}|\hat{\mathbf{z}}_t}^{-1}$, and $L_{\mathbf{z}_t|\mathbf{x}_{>t}}$ are continuous, $h$ is continuous and differentiable. Moreover, by leveraging Assumption (iii), different values of $\mathbf{z}$, i.e., $\mathbf{z}_t^{(1)}, \mathbf{z}_t^{(2}$ imply $p(\mathbf{x}_t|\mathbf{z}_t^{(1)}) \neq p(\mathbf{x}_t|\mathbf{z}_t^{(2)})$. So we can construct a function $F : \mathcal{Z} \rightarrow p(\mathbf{x}_t|\mathbf{z}_t)$, and we have:

$$\mathbf{z}_t^{(1)} \neq \mathbf{z}_t^{(2)} \longrightarrow F(\mathbf{z}_t^{(1)}) \neq F(\mathbf{z}_t^{(2)}), \tag{A23}$$

implying that $F$ is injective. Moreover, by using Equation (A22), we have $F(\mathbf{z}_t) = F(h(\hat{\mathbf{z}}_t))$, which implies $\mathbf{z}_t = h(\hat{\mathbf{z}}_t)$.

The aforementioned result leverage $\mathbf{x}_{<t}, \mathbf{x}_t$, and $\mathbf{x}_{>t}$ as three different measurement of $\mathbf{z}_t$, where $|\mathbf{x}_{<t}| \gg |\mathbf{z}_t|$, $|\mathbf{x}_{>t}| \gg |\mathbf{z}_t|$ and $|\mathbf{x}_t| < |\mathbf{z}_t|$. It may imply that when the $\mathbf{x}_t$ cannot provide enough information to recover $\mathbf{z}_t$, we can seek more information from $\mathbf{x}_{<t}$ and $\mathbf{x}_{>t}$.

Sequentially, we further prove that when the observed and $L$-layer latent variables follow the data generation process in Equation (1) and (2), we require at least $2L + 1$ adjacent observed variables, i.e., $\{\mathbf{x}_{t-L}, \cdots, \mathbf{x}_t, \cdots, \mathbf{x}_{t+L}\}$ to make $\mathbf{z}_t$ block-wise identifiable. We prove it by contradiction as follows.

Suppose we have $2L$ adjacent observations, which can be divided into two cases: 1) $\{\mathbf{x}_{t-L+1}, \cdots, \mathbf{x}_t, \cdots, \mathbf{x}_{t+L}\}$ and $\{\mathbf{x}_{t-L}, \cdots, \mathbf{x}_t, \cdots, \mathbf{x}_{t+L-1}\}$. In the first case, suppose the dimension of $\mathbf{x}_t$ and that of any layer of latent variables $\mathbf{z}_t^l$ are $n$, the dimension of $\mathbf{x}_{t-L+1}, \cdots, \mathbf{x}_{t-1}$ is $(L-1) \times n$ and the dimension of $\mathbf{z}_t$ is $L \times n$, conflicting with the assumption that $L_{\mathbf{x}_{t+1}, \cdots, \mathbf{x}_{t+L'}|\mathbf{z}_t}$ is injective. In the second case, the dimensions of $\mathbf{x}_{t-L}, \cdots, \mathbf{x}_{t-1}$ and $\mathbf{x}_{t+1}, \cdots, \mathbf{x}_{t+L-1}$ are $L \times n$ and $(L-1) \times n$, respectively, conflicting with the assumption that $L_{\mathbf{x}_{t-L'}, \cdots, \mathbf{x}_{t-1}|\mathbf{x}_{t+1}, \cdots, \mathbf{x}_{t+L}}$ is injective. As a result, we require at least $2L + 1$ adjacent observations, i.e., $\{\mathbf{x}_{t-L}, \cdots, \mathbf{x}_t, \cdots, \mathbf{x}_{t+L}\}$ to make $\mathbf{z}_t$ block-wise identifiable. $\square$

## B.3 Examples of injective linear operators

The assumption of the injectivity of a linear operator is commonly employed in the nonparametric identification [24, 3, 26]. Intuitively, it means that different input distributions of a linear operator correspond to different output distributions of that operator. For a better understanding of this assumption, we provide several examples that describe the mapping from $p_{\mathbf{a}} \rightarrow p_{\mathbf{b}}$, where $\mathbf{a}$ and $\mathbf{b}$ are random variables.

**Example 1** (**Inverse Transformation**). *$b = g(a)$, where $g$ is an invertible function.*

**Example 2** (**Additive Transformation**). *$b = a + \epsilon$, where $p(\epsilon)$ must not vanish everywhere after the Fourier transform (Theorem 2.1 in [55]).*

**Example 3.** *$b = g(a) + \epsilon$, where the same conditions from Examples 1 and 2 are required.*

**Example 4** (**Post-linear Transformation**). *$b = g_1(g_2(a)+\epsilon)$, a post-nonlinear model with invertible nonlinear functions $g_1, g_2$, combining the assumptions in **Examples 1-3**.*

**Example 5** (**Nonlinear Transformation with Exponential Family**). *$b = g(a, \epsilon)$, where the joint distribution $p(a, b)$ follows an exponential family.*

**Example 6** (**General Nonlinear Transformation**). *$b = g(a, \epsilon)$, a general nonlinear formulation. Certain deviations from the nonlinear additive model (**Example 3**), e.g., polynomial perturbations, can still be tractable.*

## B.4 Monotonicity and Normalization Assumption

**Assumption 1** (Monotonicity and Normalization Assumption [26]). *For any $\mathbf{x}_t \in \mathcal{X}_t$, there exists a known functional $G$ such that $G\left[p_{\mathbf{x}_{t+1}|\mathbf{x}_t, \mathbf{z}_t}(\cdot|\mathbf{x}_t, \mathbf{z}_t)\right]$ is monotonic in $\mathbf{z}_t$. We normalize $\mathbf{z}_t = G\left[p_{\mathbf{x}_{t+1}|\mathbf{x}_t, \mathbf{z}_t}(\cdot|\mathbf{x}_t, \mathbf{z}_t)\right]$.*

## B.5 Block-wise Identifiability of Latent Variables $\mathbf{z}_t^l$ in any $l$-th Layer

**Theorem A3.** (***Block-wise Identifiability of Latent Variables $\mathbf{z}_t^l$ in any $l$-th Layer.***) *For a series of observed variables $\mathbf{x}_t \in \mathbb{R}^n$ and estimated latent variables $\hat{\mathbf{z}}_t \in \mathbb{R}^n$ with the corresponding process $\hat{f}_i, \hat{P}(\hat{\epsilon}), \hat{P}(\hat{\eta}), \hat{\mathbf{g}}$, suppose that the process subject to observational equivalence $\mathbf{x}_t = \hat{\mathbf{g}}(\hat{\mathbf{z}}_t, \hat{\eta}_t)$. We*

let $\mathbf{c}_t \triangleq \{\mathbf{z}_{t-1}, \mathbf{z}_t\} \in \mathbb{R}^{2 \times L \times n}$ and that $\mathcal{M}_{\mathbf{c}_t}$ be the variable set of two consecutive timestamps and the corresponding node-vector Markov network respectively, and further employ following assumptions:

- *(i) (Smooth and Positive Density): The conditional probability function of the latent variables $\mathbf{c}_t$ is smooth and positive, i.e., $p(\mathbf{c}_t|\mathbf{z}_{t-2})$ is third-order differentiable and $p(\mathbf{c}_t|\mathbf{z}_{t-2}) > 0$ over $\mathbb{R}^{2 \times L \times n}$.*

- *(ii) (Sufficient Variability): Denote $|\mathcal{M}_{\mathbf{c}_t}|$ as the number of edges in Markov network $\mathcal{M}_{\mathbf{c}_t}$. Let*

$$
\begin{aligned}
w(m) = &\Big( \frac{\partial^3 \log p(\mathbf{c}_t|\mathbf{z}_{t-2})}{\partial c_{t,1}^2 \partial z_{t-2,m}}, \cdots, \frac{\partial^3 \log p(\mathbf{c}_t|\mathbf{z}_{t-2})}{\partial c_{t,2n}^2 \partial z_{t-2,m}} \Big) \oplus \\
&\Big( \frac{\partial^2 \log p(\mathbf{c}_t|\mathbf{z}_{t-2})}{\partial c_{t,1} \partial z_{t-2,m}}, \cdots, \frac{\partial^2 \log p(\mathbf{c}_t|\mathbf{z}_{t-2})}{\partial c_{t,2n} \partial z_{t-2,m}} \Big) \oplus \Big( \frac{\partial^3 \log p(\mathbf{c}_t|\mathbf{z}_{t-2})}{\partial c_{t,i} \partial c_{t,j} \partial z_{t-2,m}} \Big)_{(i,j) \in \mathcal{E}(\mathcal{M}_{\mathbf{c}_t})},
\end{aligned}
$$
$$(A24)$$

*where $\oplus$ denotes concatenation operation and $(i,j) \in \mathcal{E}(\mathcal{M}_{\mathbf{c}_t})$ denotes all pairwise indice such that $c_{t,i}, c_{t,j}$ are adjacent in $\mathcal{M}_{\mathbf{c}_t}$. For $m \in [1, \cdots, n]$, there exist $4n + |\mathcal{M}_{\mathbf{c}_t}|$ different values of $\mathbf{z}_{t-2,m}$, such that the $4n + |\mathcal{M}_{\mathbf{c}_t}|$ values of vector functions $w(m)$ are linearly independent.*

*Then for latent variables $\mathbf{z}_t^l$ at the $l$-th layer, $\mathbf{z}_t^l$ is block-wise identifiable, i.e., there exists $\hat{\mathbf{z}}_t^l$ and an invertible function $h_t^l: \mathbb{R}^n \to \mathbb{R}^n$, such that $\mathbf{z}_t^l = h_t^l(\hat{\mathbf{z}}_t^l)$.*

*Proof.* By reusing Theorem A2 with more observed variables, $\{\mathbf{z}_{t-2}, \mathbf{z}_{t-1}, \mathbf{z}_t\}$ and $\{\mathbf{z}_{t-1}, \mathbf{z}_t\}$ are also block-wise identifiable, implying that there exists invertible functions $h_3$ and $h_2$, such that $\hat{\mathbf{z}}_{t-2}, \hat{\mathbf{z}}_{t-1}, \hat{\mathbf{z}}_t = h_3(\mathbf{z}_{t-2}, \mathbf{z}_{t-1}, \mathbf{z}_t)$ and $\hat{\mathbf{z}}_{t-1}, \hat{\mathbf{z}}_t = h_2(\mathbf{z}_{t-1}, \mathbf{z}_t)$. So we have:

$$
\begin{aligned}
P(\hat{\mathbf{z}}_{t-2}, \hat{\mathbf{z}}_{t-1}, \hat{\mathbf{z}}_t) &= P(\hat{\mathbf{z}}_{t-2}, \mathbf{z}_{t-1}, \mathbf{z}_t)|\mathbf{J}_{h_2}| \iff P(\hat{\mathbf{z}}_{t-1}, \hat{\mathbf{z}}_t|\hat{\mathbf{z}}_{t-2}) = P(\mathbf{z}_{t-1}, \mathbf{z}_t|\hat{\mathbf{z}}_{t-2})|\mathbf{J}_{h_2}| \\
&\iff P(\hat{\mathbf{c}}_t|\hat{\mathbf{z}}_{t-2}) = P(\mathbf{c}_t|\hat{\mathbf{z}}_{t-2})|\mathbf{J}_{h_2}| \iff \ln P(\hat{\mathbf{c}}_t|\hat{\mathbf{z}}_{t-2}) = \ln P(\mathbf{c}_t|\hat{\mathbf{z}}_{t-2}) + \ln |\mathbf{J}_{h_2}|,
\end{aligned}
$$
$$(A25)$$

And then we partition the latent variables $\mathbf{z}_t$ into two parts $\mathbf{z}_t$

Let $\hat{\mathbf{z}}_{t-1,k}^{l_1}$ and $\hat{\mathbf{z}}_{t,o}^{l_2}$ be two variables that denote the $k$-th variable of $\mathbf{z}_{t-1}^{l_1}$ and $o$-th variable of $\mathbf{z}_t^{l_2}$, respectively, where $l_1 > l_2$. According to the data generation process, it is not hard to find that $\hat{\mathbf{z}}_{t-1,k}^{l_1}$ and $\hat{\mathbf{z}}_{t,o}^{l_2}$ are not adjacent in the estimated Markov network $\mathcal{M}_{\hat{\mathbf{c}}_t}$ over $\hat{\mathbf{c}}_t = \{\hat{\mathbf{z}}_{t-1}, \hat{\mathbf{z}}_t\}$. We conduct the first-order derivative w.r.t. $\hat{\mathbf{z}}_{t-1,k}^{l_1}$ and have

$$
\frac{\partial \log p(\hat{\mathbf{c}}_t|\hat{\mathbf{z}}_{t-2})}{\partial \hat{\mathbf{z}}_{t-1,k}^{l_1}} = \sum_{l=1}^{2L} \sum_{i=1}^{n} \frac{\partial \log p(\mathbf{c}_t|\hat{\mathbf{z}}_{t-2})}{\partial c_{t,i}^l} \cdot \frac{\partial c_{t,i}^l}{\partial \hat{\mathbf{z}}_{t-1,k}^{l_1}} + \frac{\partial \log |\mathbf{J}_{h_2}|}{\partial \hat{\mathbf{z}}_{t-1,k}^{l_1}}.
\tag{A26}
$$

We further conduct the second-order derivative w.r.t. $\hat{\mathbf{z}}_{t-1,k}^{l_1}$ and $\hat{\mathbf{z}}_{t,o}^{l_2}$, then we have:

$$
\begin{aligned}
\frac{\partial^2 \log p(\hat{\mathbf{c}}_t|\hat{\mathbf{z}}_{t-2})}{\partial \hat{\mathbf{z}}_{t-1,k}^{l_1} \partial \hat{\mathbf{z}}_{t,o}^{l_2}} &= \sum_{l=1}^{2L} \sum_{i=1}^{n} \sum_{s=1}^{2L} \sum_{j=1}^{n} \frac{\partial^2 \log p(\mathbf{c}_t|\hat{\mathbf{z}}_{t-2})}{\partial c_{t,i}^l \partial c_{t,j}^s} \cdot \frac{\partial c_{t,i}^l}{\partial \hat{\mathbf{z}}_{t-1,k}^{l_1}} \cdot \frac{\partial c_{t,j}^s}{\partial \hat{\mathbf{z}}_{t,o}^{l_2}} \\
&+ \sum_{l=1}^{2L} \sum_{i=1}^{n} \frac{\partial \log p(\mathbf{c}_t|\hat{\mathbf{z}}_{t-2})}{\partial c_{t,i}^l} \cdot \frac{\partial^2 c_{t,i}^l}{\partial \hat{\mathbf{z}}_{t-1,k}^{l_1} \partial \hat{\mathbf{z}}_{t,o}^{l_2}} + \frac{\partial^2 \log |\mathbf{J}_{h_2}|}{\partial \hat{\mathbf{z}}_{t-1,k}^{l_1} \partial \hat{\mathbf{z}}_{t,o}^{l_2}}.
\end{aligned}
\tag{A27}
$$

Since $\hat{\mathbf{z}}_{t-1,k}^{l_1}$ and $\hat{\mathbf{z}}_{t,o}^{l_2}$ are not adjacent in $\mathcal{M}_{\hat{\mathbf{c}}_t}$, $\hat{\mathbf{z}}_{t-1,k}^{l_1}$ and $\hat{\mathbf{z}}_{t,o}^{l_2}$ are conditionally independent given $\hat{\mathbf{c}}_t \backslash \{\hat{c}_{t,k}, \hat{c}_{t,l}\}$. Utilizing the fact that conditional independence can lead to zero cross derivative [53], for each value of $\hat{\mathbf{z}}_{t-2}$, we have:

$$
\begin{aligned}
\frac{\partial^2 \log p(\hat{\mathbf{c}}_t|\hat{\mathbf{z}}_{t-2})}{\partial \hat{\mathbf{z}}_{t-1,k}^{l_1} \partial \hat{\mathbf{z}}_{t,o}^{l_2}} &= \frac{\partial^2 \log p(\hat{c}_{t,k}|\hat{\mathbf{c}}_t \backslash \{\hat{\mathbf{z}}_{t-1,k}^{l_1}, \hat{\mathbf{z}}_{t,o}^{l_2}\}, \hat{\mathbf{z}}_{t-2})}{\partial \hat{\mathbf{z}}_{t-1,k}^{l_1} \partial \hat{\mathbf{z}}_{t,o}^{l_2}} + \frac{\partial^2 \log p(\hat{c}_{t,l}|\mathbf{c}_t \backslash \{\hat{\mathbf{z}}_{t-1,k}^{l_1}, \hat{\mathbf{z}}_{t,o}^{l_2}\}, \hat{\mathbf{z}}_{t-2})}{\partial \hat{\mathbf{z}}_{t-1,k}^{l_1} \partial \hat{\mathbf{z}}_{t,o}^{l_2}} \\
&+ \frac{\partial^2 \log p(\hat{\mathbf{c}}_t \backslash \{\hat{\mathbf{z}}_{t-1,k}^{l_1}, \hat{\mathbf{z}}_{t,o}^{l_2}\}|\hat{\mathbf{z}}_{t-2})}{\partial \hat{\mathbf{z}}_{t-1,k}^{l_1} \partial \hat{\mathbf{z}}_{t,o}^{l_2}} = 0.
\end{aligned}
$$
$$(A28)$$

Bring in Equation (A28), Equation (A27) can be further derived as:

$$
0 = \underbrace{\sum_{l=1}^{2L}\sum_{i=1}^{n} \frac{\partial^2 \log p(\mathbf{c}_t|\hat{\mathbf{z}}_{t-2})}{\partial (c_{t,i}^l)^2} \cdot \frac{\partial c_{t,i}^l}{\partial \hat{\mathbf{z}}_{t-1,k}^{l_1}} \cdot \frac{\partial c_{t,i}^l}{\partial \hat{\mathbf{z}}_{t,o}^{l_2}}}_{\textbf{(i) } i=j}
$$

$$
+ \underbrace{\sum_{l=1}^{2L}\sum_{i=1}^{n} \sum_{s:(s,l)\in\mathcal{E}(\mathcal{M}_{\mathbf{c}_t})}\sum_{j=1}^{n} \frac{\partial^2 \log p(\mathbf{c}_t|\hat{\mathbf{z}}_{t-2})}{\partial c_{t,i}^l \partial c_{t,j}^s} \cdot \frac{\partial c_{t,i}^l}{\partial \hat{\mathbf{z}}_{t-1,k}^{l_1}} \cdot \frac{\partial c_{t,j}^s}{\partial \hat{\mathbf{z}}_{t,o}^{l_2}}}_{\textbf{(ii)}c_{t,i} \text{ and } c_{t,j} \text{ are adjacent in } \mathcal{M}_{\mathbf{c}_t}} \tag{A29}
$$

$$
+ \underbrace{\sum_{l=1}^{2L}\sum_{i=1}^{n} \sum_{s:(s,l)\notin\mathcal{E}(\mathcal{M}_{\mathbf{c}_t})}\sum_{j=1}^{n} \frac{\partial^2 \log p(\mathbf{c}_t|\hat{\mathbf{z}}_{t-2})}{\partial c_{t,i}^l \partial c_{t,j}^s} \cdot \frac{\partial c_{t,i}^l}{\partial \hat{\mathbf{z}}_{t-1,k}^{l_1}} \cdot \frac{\partial c_{t,j}^s}{\partial \hat{\mathbf{z}}_{t,o}^{l_2}}}_{\textbf{(iii)}c_{t,i} \text{ and } c_{t,j} \text{ are \textbf{not} adjacent in } \mathcal{M}_{\mathbf{c}_t}}
$$

$$
+ \sum_{l=1}^{2L}\sum_{i=1}^{n} \frac{\partial \log p(\mathbf{c}_t|\hat{\mathbf{z}}_{t-2})}{\partial c_{t,i}^l} \cdot \frac{\partial^2 c_{t,i}^l}{\partial \hat{\mathbf{z}}_{t-1,k}^{l_1}\partial \hat{\mathbf{z}}_{t,o}^{l_2}} + \frac{\partial \log |\mathbf{J}_{h_c,t}|}{\partial \hat{\mathbf{z}}_{t-1,k}^{l_1}\partial \hat{\mathbf{z}}_{t,o}^{l_2}},
$$

where $(j,i) \in \mathcal{E}(\mathcal{M}_{\mathbf{c}_t})$ denotes that $c_{t,i}$ and $c_{t,j}$ are adjacent in $\mathcal{M}_{\mathbf{c}_t}$. Similar to Equation (A28), we have $\frac{\partial^2 p(\mathbf{c}_t|\mathbf{z}_{t-2})}{\partial c_{t,i}\partial c_{t,j}} = 0$ when $c_{t,i}, c_{t,j}$ are not adjacent in $\mathcal{M}_{\mathbf{c}_t}$. Thus, Equation (A29) can be rewritten as:

$$
0 = \sum_{l=1}^{2L}\sum_{i=1}^{n} \frac{\partial^2 \log p(\mathbf{c}_t|\hat{\mathbf{z}}_{t-2})}{\partial (c_{t,i}^l)^2} \cdot \frac{\partial c_{t,i}^l}{\partial \hat{\mathbf{z}}_{t-1,k}^{l_1}} \cdot \frac{\partial c_{t,i}^l}{\partial \hat{\mathbf{z}}_{t,o}^{l_2}} + \sum_{l=1}^{2L}\sum_{i=1}^{n} \sum_{s:(s,l)\in\mathcal{E}(\mathcal{M}_{\mathbf{c}_t})}\sum_{j=1}^{n} \frac{\partial^2 \log p(\mathbf{c}_t|\hat{\mathbf{z}}_{t-2})}{\partial c_{t,i}^l \partial c_{t,j}^s} \cdot \frac{\partial c_{t,i}^l}{\partial \hat{\mathbf{z}}_{t-1,k}^{l_1}} \cdot \frac{\partial c_{t,j}^s}{\partial \hat{\mathbf{z}}_{t,o}^{l_2}}
$$

$$
+ \sum_{l=1}^{2L}\sum_{i=1}^{n} \frac{\partial \log p(\mathbf{c}_t|\hat{\mathbf{z}}_{t-2})}{\partial c_{t,i}^l} \cdot \frac{\partial^2 c_{t,i}^l}{\partial \hat{\mathbf{z}}_{t-1,k}^{l_1}\partial \hat{\mathbf{z}}_{t,o}^{l_2}} + \frac{\partial \log |\mathbf{J}_{h_c,t}|}{\partial \hat{\mathbf{z}}_{t-1,k}^{l_1}\partial \hat{\mathbf{z}}_{t,o}^{l_2}},
\tag{A30}
$$

Then for each $m = 1, 2, \cdots, nL$ and each value of $\hat{\mathbf{z}}_{t-2,m}$, we conduct partial derivative on both sides of Equation (A30) and have:

$$
0 = \sum_{l=1}^{2L}\sum_{i=1}^{n} \frac{\partial^3 \log p(\mathbf{c}_t|\hat{\mathbf{z}}_{t-2})}{\partial (c_{t,i}^l)^2 \partial \hat{\mathbf{z}}_{t-2,m}} \cdot \frac{\partial c_{t,i}^l}{\partial \hat{\mathbf{z}}_{t-1,k}^{l_1}} \cdot \frac{\partial c_{t,i}^l}{\partial \hat{\mathbf{z}}_{t,o}^{l_2}} +
$$

$$
\sum_{l=1}^{2L}\sum_{i=1}^{n} \sum_{s:(s,l)\in\mathcal{E}(\mathcal{M}_{\mathbf{c}})}\sum_{j=1}^{n} \frac{\partial^3 \log p(\mathbf{c}_t|\hat{\mathbf{z}}_{t-2})}{\partial c_{t,i}^l \partial c_{t,j}^s \partial \hat{\mathbf{z}}_{t-2,m}} \cdot \frac{\partial c_{t,i}^l}{\partial \hat{\mathbf{z}}_{t-1,k}^{l_1}} \cdot \frac{\partial c_{t,j}^s}{\hat{\mathbf{z}}_{t,o}^{l_2}} \tag{A31}
$$

$$
+ \sum_{l=1}^{2L}\sum_{i=1}^{n} \frac{\partial^2 \log p(c_t|\hat{\mathbf{z}}_{t-2})}{\partial c_{t,i}^l \partial \hat{\mathbf{z}}_{t-2,m}} \cdot \frac{\partial^2 c_{t,i}^l}{\partial \hat{\mathbf{z}}_{t-1,k}^{l_1}\partial \hat{\mathbf{z}}_{t,o}^{l_2}},
$$

Finally, we have

$$
0 = \sum_{l=1}^{2L}\sum_{i=1}^{n} \frac{\partial^3 \log p(\mathbf{c}_t|\hat{\mathbf{z}}_{t-2})}{\partial (c_{t,i}^l)^2 \partial \hat{\mathbf{z}}_{t-2,m}} \cdot \frac{\partial c_{t,i}^l}{\partial \hat{\mathbf{z}}_{t-1,k}^{l_1}} \cdot \frac{\partial c_{t,i}^l}{\partial \hat{\mathbf{z}}_{t,o}^{l_2}} + \sum_{l=1}^{2L}\sum_{i=1}^{n} \frac{\partial^2 \log p(c_t|\hat{\mathbf{z}}_{t-2})}{\partial c_{t,i}^l \partial \hat{\mathbf{z}}_{t-2,m}} \cdot \frac{\partial^2 c_{t,i}^l}{\partial \hat{\mathbf{z}}_{t-1,k}^{l_1}\partial \hat{\mathbf{z}}_{t,o}^{l_2}}
$$

$$
+ \sum_{l,s:(l,s)\in\mathcal{E}(\mathcal{M}_{\mathbf{c}})}\sum_{i=1}^{n}\sum_{j=1}^{n} \frac{\partial^3 \log p(\mathbf{c}_t|\hat{\mathbf{z}}_{t-2})}{\partial c_{t,i}^l \partial c_{t,j}^s \partial \hat{\mathbf{z}}_{t-2,m}} \cdot \left( \frac{\partial c_{t,i}^l}{\partial \hat{\mathbf{z}}_{t-1,k}^{l_1}} \cdot \frac{\partial c_{t,j}^s}{\partial \hat{\mathbf{z}}_{t,o}^{l_2}} + \frac{\partial c_{t,j}^s}{\partial \hat{\mathbf{z}}_{t-1,k}^{l_1}} \cdot \frac{\partial c_{t,i}^l}{\partial \hat{\mathbf{z}}_{t,o}^{l_2}} \right). \tag{A32}
$$

According to Assumption A2, we can construct $4nL + |\mathcal{M}_{\mathbf{c}}| \times n^2$ different equations with different values of $\hat{\mathbf{z}}_{t-2,m}$, and the coefficients of the equation system they form are linearly independent. To ensure that the right-hand side of the equations is always 0, the only solution is

$$
\frac{\partial c_{t,i}^l}{\partial \hat{\mathbf{z}}_{t-1,k}^{l_1}} \cdot \frac{\partial c_{t,i}^l}{\partial \hat{\mathbf{z}}_{t,o}^{l_2}} = 0, \tag{A33}
$$

$$\frac{\partial c_{t,i}^l}{\partial \hat{\mathbf{z}}_{t-1,k}^{l_1}} \cdot \frac{\partial c_{t,j}^s}{\partial \hat{\mathbf{z}}_{t,o}^{l_2}} + \frac{\partial c_{t,j}^s}{\partial \hat{\mathbf{z}}_{t-1,k}^{l_1}} \cdot \frac{\partial c_{t,i}^l}{\partial \hat{\mathbf{z}}_{t,o}^{l_2}} = 0, \tag{A34}$$

$$\frac{\partial (c_{t,i}^l)^2}{\partial \hat{\mathbf{z}}_{t-1,k}^{l_1} \partial \hat{\mathbf{z}}_{t,o}^{l_2}} = 0. \tag{A35}$$

Bringing Eq A33 into Eq A34, at least one product must be zero, and the other must be zero as well. That is,

$$\frac{\partial c_{t,i}^l}{\partial \hat{\mathbf{z}}_{t-1,k}^{l_1}} \cdot \frac{\partial c_{t,j}^s}{\partial \hat{\mathbf{z}}_{t,o}^{l_2}} = 0. \tag{A36}$$

According to the aforementioned results, $\hat{\mathbf{z}}_{t-1,k}^{l_1}$ and $\hat{\mathbf{z}}_{t,o}^{l_2}$ be two variables that denote the $k$-th variable of $\mathbf{z}_{t-1}^{l_1}$ and $o$-th variable of $\mathbf{z}_t^{l_2}$, respectively, where $l_1 > l_2$, we draw the following conclusions.
**(i)** Equation (A33) implies that, each ground-truth latent variable $c_{t,i}^l$ in $l$-th block is a function of at most one of $\hat{c}_{t,k}^{l_1}$ and $\hat{c}_{t,l}^{l_2}$, which are in $l_1$-th and $l_2$-th layer, respectively
**(ii)** Equation (A36) implies that, for each pair of ground-truth latent variables $c_{t,i}^l$ and $c_{t,j}^s$ in $l$-th and $l$-th blocks, respectively, that are **adjacent** in $\mathcal{M}_{\mathbf{c}_t}$ over $\mathbf{c}_t$, they can not be a function of $\hat{c}_{t,k}^{l_1}$ and $\hat{c}_{t,l}^{l_2}$ respectively.

According to the data generation process, we can restrict the independent noises among blocks, and hence the estimated node-vector Markov network is isomorphic to the ground-truth node-vector Markov network.

Sequentially, we further give the proof that under the same permutation $\mathbf{z}_t^\pi$, block $z_t^i$ is only a function of $z_t^{\pi(i)}$. Since the permutation happens on each timestamp respectively, the cross-timestamp block-wise identifiability is presented clearly.

Suppose there exists a pair of indices $l, s \in \{1, \cdots, L\}$. Since $h_2$ is invertible, there exists a permuted version of the estimated blocks, denoted as $\mathbf{c}_t^\pi$, such that:

$$\frac{\partial \mathbf{z}_{t,i}^l}{\partial \hat{\mathbf{z}}_{t,j}^{\pi(l)}} \neq 0, \quad l = 1, \cdots, L, \text{ and } i, j = 1, \cdots, n, \tag{A37}$$

we have $\frac{\partial \mathbf{z}_{t,i}^l}{\partial \hat{\mathbf{z}}_{t,j}^{\pi(l)}} \neq 0$ and $\frac{\partial \mathbf{z}_{t,i}^s}{\partial \hat{\mathbf{z}}_{t,j}^{\pi(s)}} \neq 0$. Let us discuss it case by case. We can discuss it in the following case.

- If $\mathbf{z}_t^l$ is not adjacent to $\mathbf{z}_t^s$, we have $\hat{\mathbf{z}}_t^{\pi(l)}$ is not adjacent to $\hat{\mathbf{z}}_t^{\pi(s)}$. Using Equation (A33), we have $\frac{\partial \mathbf{c}_{t,i}^l}{\partial \hat{\mathbf{z}}_{t-1,k}^{l_1}} \cdot \frac{\partial \mathbf{c}_{t,i}^l}{\partial \hat{\mathbf{z}}_{t,o}^{l_2}} = 0$, which leads to $\frac{\partial \mathbf{z}_t^l}{\partial \hat{\mathbf{z}}_t^{\pi(j)}} = 0$.

- If block $\mathbf{z}_t^l$ is adjacent to block $\mathbf{z}_t^s$, we have $\hat{\mathbf{z}}_t^{\pi(i)}$ is adjacent to $\hat{\mathbf{z}}_t^{\pi(j)}$. The intimate neighbor set of $\mathbf{z}_t^i$ is empty, there exists at least one index $k$ such that $\mathbf{z}_t^k$ is adjacent to $\mathbf{z}_t^i$ but not adjacent to $\mathbf{z}_t^j$. Similarly, we have the same structure on the estimated Markov network, which means that $\hat{\mathbf{z}}_t^{\pi(k)}$ is adjacent to $\hat{\mathbf{z}}_t^{\pi(i)}$ but not adjacent to $\hat{\mathbf{z}}_t^{\pi(j)}$. Using Equation (A36) we have $\frac{\partial \mathbf{c}_{t,i}^l}{\partial \hat{\mathbf{z}}_{t-1,k}^{l_1}} \cdot \frac{\partial \mathbf{c}_{t,j}^s}{\partial \hat{\mathbf{z}}_{t,o}^{l_2}} = 0$, which leads to $\frac{\partial \mathbf{z}_{t,i}^l}{\partial \hat{\mathbf{z}}_{t,j}^{\pi(s)}} = 0$.

In conclusion, we always have $\frac{\partial \mathbf{z}_{t,i}^l}{\partial \hat{\mathbf{z}}_{t,j}^{\pi(s)}} = 0$, meaning that there exists a permutation $\pi$ of the estimated blocks, such that $\mathbf{z}_t^l$ and $\mathbf{z}_t^{\pi(l)}$ is one-to-one corresponding.

Sequentially, we further leverage Lemma A2 to show that there is exist an invertible function $h^l$, such that $\hat{\mathbf{z}}_t^l = h^l(\mathbf{z}_t^l)$. $\qquad\square$

**Lemma A2.** *(Hierarchical Structure Resolves Layer-Wise Permutation Indeterminacy) If the estimated latent causal graph represents a hierarchical structure, the layer-wise components $\mathbf{z}_t^1, \mathbf{z}_t^2, \cdots, \mathbf{z}_t^L$ are block-wise identifiable without permutation.*

*Proof.* Let $\mathbf{A}_t$ represent the true causal adjacency matrix of the latent causal graph at time $t$ in layer-wise level, and let $\hat{\mathbf{A}}_t$ represent its estimation. By definition, we have

$$\mathbf{A}_{t,k,l} = \begin{cases} \frac{\partial \mathbf{z}_t^l}{\partial \mathbf{z}_t^k}, & k = l+1, \\ 0, & \text{otherwise}, \end{cases} \quad \hat{\mathbf{A}}_{t,k,l} = \begin{cases} \frac{\partial \hat{\mathbf{z}}_t^l}{\partial \hat{\mathbf{z}}_t^k}, & k = l+1, \\ 0, & \text{otherwise}, \end{cases} \quad k, l = 1, \dots, L. \tag{A38}$$

Consider the nonzero elements of the adjacency matrices, which occur at positions where $k = l + 1$. If a layer-wise permutation is applied, the rows and columns of the $\mathbf{A}_t$ are both permuted according to the same permutation. Specifically, we have

$$\hat{\mathbf{A}}_{l+1,l} = \mathbf{D}_{l+1,1} \mathbf{A}_{\pi(l+1),\pi(l)}, \tag{A39}$$

where $\mathbf{D}_{l+1,1}$ is a diagonal matrix representing the scaling indeterminacy, and $\pi : \{1, \dots, L\} \rightarrow \{1, \dots, L\}$ is a permutation function. The subscripts of $\hat{\mathbf{A}}$ and $\mathbf{A}$ above indicate the following equation:

$$\pi(l+1) = \pi(l) + 1. \tag{A40}$$

The recursive formula in Equation (A40) implies:

$$\pi(l) = \pi(1) + (l - 1). \tag{A41}$$

For $\pi(l)$ to be a valid permutation covering all values in $\{1, \dots, L\}$, it is necessary that $\pi(1) = 1$. If $\pi(1) \neq 1$, the sequence $\pi(l) = \pi(1) + (l - 1)$ would either exceed $L$ or fail to include 1, violating the bijective property of $\pi$. Hence, we conclude that

$$\pi(l) = l, \tag{A42}$$

indicating that the layer-wise components remain unpermuted. $\qquad \square$

## B.6 Extension to Multiple Lags

For the sake of simplicity, we consider only one special case with $\tau = 1$ in Theorem 2. Our theoretical results can actually be extended to arbitrary lags and subsequences easily. For any given time lag $\tau$, and future horizons which is centered at $\mathbf{z}_t$ with historical $\tau_h$ and future $\tau_f$ steps, i.e., $\mathbf{c}_t = \{\mathbf{z}_{t-\tau_h}, \cdots, \mathbf{z}_t, \cdots, \mathbf{z}_{t+\tau_f}\} \in \mathbb{R}^{(\tau_h + \tau_f + n) \times n}$. In this case, the vector function $w(m)$ in the Sufficient Variability assumption should be modified as

$$
\begin{aligned}
w(m) = & \left( \frac{\partial^3 \log p(\mathbf{c}_t | \mathbf{z}_{t-\tau_h-1}, \cdots, \mathbf{z}_{t-\tau_h-\tau})}{\partial c_{t,1}^2 \partial z_{t-\tau_h-1,m}}, \cdots, \frac{\partial^3 \log p(\mathbf{c}_t | \mathbf{z}_{t-\tau_h-1}, \cdots, \mathbf{z}_{t-\tau_h-\tau})}{\partial c_{t,2n}^2 \partial z_{t-\tau_h-1,m}} \right) \oplus \\
& \left( \frac{\partial^2 \log p(\mathbf{c}_t | \mathbf{z}_{t-\tau_h-1}, \cdots, \mathbf{z}_{t-\tau_h-\tau})}{\partial c_{t,1} \partial z_{t-\tau_h-1,m}}, \cdots, \frac{\partial^2 \log p(\mathbf{c}_t | \mathbf{z}_{t-\tau_h-1}, \cdots, \mathbf{z}_{t-\tau_h-\tau})}{\partial c_{t,2n} \partial z_{t-\tau_h-1,m}} \right) \oplus \\
& \left( \frac{\partial^3 \log p(\mathbf{c}_t | \mathbf{z}_{t-\tau_h-1}, \cdots, \mathbf{z}_{t-\tau_h-\tau})}{\partial c_{t,i} \partial c_{t,j} \partial z_{t-\tau_h-1,m}} \right)_{(i,j) \in \mathcal{E}(\mathcal{M}_{\mathbf{c}_t})}.
\end{aligned} \tag{A43}
$$

Besides, $2 \times n \times (\tau_h + \tau_f + 1) + |\mathcal{M}_{\mathbf{c}_t}|$ values of linearly independent vector functions in $z_{t',m}^l$ for $t' \in \{t - \tau_h - 1, \cdots, t - \tau_h - \tau\}$ and $m \in \{1, \cdots, n\}$ are required as well. The rest of the theorem remains the same, and the proof can be easily extended in such a setting.

## B.7 Component-wise Identifiability of Latent Variables $z_{t,i}^l$ in any $l$-th Layer

**Lemma A3.** *(Component-wise Identifiability of Latent Variables $z_{t,i}^l$ in any $l$-th Layer) For a series of observed variables $\mathbf{x}_t \in \mathbb{R}^n$ and estimated latent variables $\hat{\mathbf{z}}_t \in \mathbb{R}^n$ with the corresponding process $\hat{f}_i, \hat{P}(\hat{\epsilon}), \hat{P}(\hat{\eta}), \hat{\mathbf{g}}$, suppose that the process subject to observational equivalence $\mathbf{x}_t = \hat{\mathbf{g}}(\hat{\mathbf{z}}_t, \hat{\eta}_t)$. We employ sufficient variability assumptions as follows:*

- *(i) (sufficient variability) For any $\mathbf{z}_t^l \in \mathcal{Z}_t^l \subseteq \mathbb{R}^n$ and $\hat{\mathbf{u}} = \{\hat{\mathbf{z}}_{t-1}^l, \hat{\mathbf{z}}_t^{l+1}\}$ there exist $2n + 1$ different values of $\hat{\mathbf{u}}$, $m = 0, \cdots, 2n$, such that these $2n$ vectors $\mathbf{v}_{l,m}$ - $\mathbf{v}_{l,0}$ are linearly independent, where $\mathbf{v}_{l,m}$ is defined as:*

$$\mathbf{v}_{l,m} = \left( \frac{\partial^2 \ln P(\mathbf{z}_{t,1}^l | \hat{\mathbf{u}})}{(\partial \mathbf{z}_{t,1}^l)^2}, \cdots, \frac{\partial^2 \ln P(\mathbf{z}_{t,n}^l | \hat{\mathbf{u}})}{(\partial \mathbf{z}_{t,n}^l)^2}, \frac{\partial \ln P(\mathbf{z}_{t,1}^l | \hat{\mathbf{u}})}{\partial \mathbf{z}_{t,1}^l}, \cdots, \frac{\partial \ln P(\mathbf{z}_{t,n}^l | \hat{\mathbf{u}})}{\partial \mathbf{z}_{t,n}^l} \right). \tag{A44}$$

*Then for $i$-th latent variable $\mathbf{z}_{t,i}^l$ at the $l-$th layer, $\mathbf{z}_{t,i}^l$ is component-wise identifiability, i.e., there exists $\hat{\mathbf{z}}_{t,j}^l$ and an invertible function $h_t^l : \mathbb{R} \to \mathbb{R}$, such that $\hat{\mathbf{z}}_{t,j}^l = h_t^l(\mathbf{z}_{t,i}^l)$.*

*Proof.* According to Theorem A3, we have achieved the block-identifiability of $\mathbf{z}_t^l$, by letting $\mathbf{z}_{t-1}^l$ and $\mathbf{z}_t^{l-1}$ be the temporal and hierarchical parents, we further have :

$$P(\hat{\mathbf{z}}_t^l, \hat{\mathbf{z}}_{t-1}^l, \hat{\mathbf{z}}_t^{l-1}) = P(h_t^l(\mathbf{z}_t^l), \hat{\mathbf{z}}_{t-1}^l, \hat{\mathbf{z}}_t^{l-1}) \iff P(\hat{\mathbf{z}}_t | \hat{\mathbf{z}}_{t-1}^l, \hat{\mathbf{z}}_t^{l-1}) = P(h_t^l(\mathbf{z}_t^l) | \hat{\mathbf{z}}_{t-1}^l, \hat{\mathbf{z}}_t^{l-1})$$

$$\iff \ln P(\hat{\mathbf{z}}_t^l | \hat{\mathbf{u}}) = \ln P(\mathbf{z}_t^l | \hat{\mathbf{u}}) + \ln |\mathbf{J}_{h_t^l}| \iff \sum_{i=1}^n \ln P(\hat{\mathbf{z}}_{t,i}^l | \hat{\mathbf{u}}) = \sum_{i=1}^n \ln P(\mathbf{z}_{t,i}^l | \hat{\mathbf{u}}) + \ln |\mathbf{J}_{h_t^l}|,$$

(A45)

where $\mathbf{J}_{h_t^l}$ is the Jacobian matrix of the transformation associated with $h_t^l$. Sequentially, we differentiate both sides of the Equation (A45) w.r.t $\hat{\mathbf{z}}_{t,j}^l$ and $\hat{\mathbf{z}}_{t,k}^l$, where $j, k \in \{1, \cdots, n\}$ and $j \neq k$ yields

$$0 = \sum_{i=1}^n \left( \frac{\partial^2 \ln P(\mathbf{z}_{t,i}^l | \hat{\mathbf{u}})}{(\partial \mathbf{z}_{t,i}^l)^2} \cdot \frac{\partial \mathbf{z}_{t,i}^l}{\partial \hat{\mathbf{z}}_{t,j}^l} \cdot \frac{\partial \mathbf{z}_{t,i}^l}{\partial \hat{\mathbf{z}}_{t,k}^l} + \frac{\partial \ln P(\mathbf{z}_{t,i}^l | \hat{\mathbf{u}})}{\partial \mathbf{z}_{t,i}^l} \cdot \frac{\partial^2 \mathbf{z}_{t,i}^l}{\partial \hat{\mathbf{z}}_{t,j}^l \partial \hat{\mathbf{z}}_{t,k}^l} \right) + \frac{\partial |\mathbf{J}_{h_t^l}|}{\partial \hat{\mathbf{z}}_{t,j}^l \partial \hat{\mathbf{z}}_{t,k}^l}. \quad \text{(A46)}$$

Therefore, for $\mathbf{u} = \mathbf{u}_0, \cdots, \mathbf{u}_{2n}$, we have $2n + 1$ such equations. Subtracting each equation corresponding to $\mathbf{u}_1, \cdots, \mathbf{u}_{2n}$ with the equation corresponding to $\mathbf{u}_0$ results in $2n$ equations:

$$0 = \sum_{i=1}^n \left( \left( \frac{\partial^2 \ln P(\mathbf{z}_{t,i}^l | \hat{\mathbf{u}}_m)}{(\partial \mathbf{z}_{t,i}^l)^2} - \frac{\partial^2 \ln P(\mathbf{z}_{t,i}^l | \hat{\mathbf{u}}_0)}{(\partial \mathbf{z}_{t,i}^l)^2} \right) \cdot \frac{\partial \mathbf{z}_{t,i}^l}{\partial \hat{\mathbf{z}}_{t,j}^l} \cdot \frac{\partial \mathbf{z}_{t,i}^l}{\partial \hat{\mathbf{z}}_{t,k}^l} \right.$$
$$\left. + \left( \frac{\partial \ln P(\mathbf{z}_{t,i}^l | \hat{\mathbf{u}}_m)}{\partial \mathbf{z}_{t,i}^l} - \frac{\partial \ln P(\mathbf{z}_{t,i}^l | \hat{\mathbf{u}}_0)}{\partial \mathbf{z}_{t,i}^l} \right) \cdot \frac{\partial^2 \mathbf{z}_{t,i}^l}{\partial \hat{\mathbf{z}}_{t,j}^l \partial \hat{\mathbf{z}}_{t,k}^l} \right),$$

(A47)

where $m = 1, \cdots, 2n$. As a result, under the sufficient variability assumption, the linear system is a $2n \times 2n$ full-rank system, and the only solution is $\frac{\partial \mathbf{z}_{t,i}^l}{\partial \hat{\mathbf{z}}_{t,j}^l} \cdot \frac{\partial \mathbf{z}_{t,i}^l}{\partial \hat{\mathbf{z}}_{t,k}^l} = 0$ and $\frac{\partial^2 \mathbf{z}_{t,i}^l}{\partial \hat{\mathbf{z}}_{t,j}^l \partial \hat{\mathbf{z}}_{t,k}^l}$. And $\frac{\partial \mathbf{z}_{t,i}^l}{\partial \hat{\mathbf{z}}_{t,j}^l} \cdot \frac{\partial \mathbf{z}_{t,i}^l}{\partial \hat{\mathbf{z}}_{t,k}^l} = 0$ implies that for each $i = 1, \cdots, n$, $\frac{\partial \mathbf{z}_{t,i}^l}{\partial \hat{\mathbf{z}}_{t,j}^l} \neq 0$ for at most one element $j \in \{1, \cdots, n\}$. Therefore, there is only at most one non-zero entry in each row indexed by $i$ in the Jacobian $\mathbf{J}_{h_t^l}$. Moreover, the invertibility of $h_t^l$ necessitates $\mathbf{J}_{h_t^l}$ to be full-rank which implies that there is exactly one non-zero component in each row of $\mathbf{J}_{h_t^l}$, implying that the $\mathbf{z}_{t,i}^l$ is component-wise identifiable. $\qquad \square$

## C  Real-world Implications of Modeling Time Series Data in a Hierarchical Manner

Real-world scenarios are usually governed by hierarchical layers of temporal latent dynamics instead of a single one. Capturing the hierarchical temporal latent dynamics plays a critical role in modeling time series data. Here is two real-world scenarios.

**Human Motion Modeling.**   Human locomotion unfolds over multiple temporal layers [16, 60, 2]:

- **Higher-level latent variables:** Capture coarse movement categories—e.g., walking versus running or resting—capturing the switch-like changes in overall gait dynamics.

- **Lower-level latent variables:** Resolve fine-grained kinematics such as joint angles, stride frequency, and instantaneous speed.

This decomposition isolates semantically meaningful factors: the high-level code tells what activity is occurring, while the low-level code details how it is executed. Such a structure supports downstream tasks, including activity recognition, early anomaly detection (e.g., pathological gaits), and personalized coaching systems that adapt exercises to an individual's movement profile.

**Climate Modeling.** Atmospheric processes also exhibit a natural hierarchy [34, 71], which is shown as follows:

- **Top-level latent variables** capture large-scale regimes like monsoon cycles or transitions between seasons, governing the dominant energy balance of a region.
- **Middle-level latent variables** reflect regional or intra-seasonal patterns—e.g., the twelve traditional solar terms in East Asia or monthly precipitation phases that modulate local ecosystems.
- **Bottom-level latent variables** track short-term fluctuations in temperature, humidity, and wind that drive daily weather variability.

Modeling these tiers jointly enables scientists to disentangle long-term trends from transient disturbances, improving extended-range forecasts, pinpointing abnormal events (heat-wave onsets, unseasonal cold snaps), and facilitating attribution studies of climate change.

## D   Relationship between Time Series Generation and Hierarchical Dynamics

The goal of time series generation is to identify the joint distribution of observed variables of different time steps, i.e., $p(\mathbf{x}_1, \cdots, \mathbf{x}_T)$. To achieve this goal, we should simultaneously capture long-range structure spanning hundreds of steps and fine-grained, moment-to-moment variability. Collapsing every timescale into a single latent layer forces one hidden state to shoulder both roles. This entangles long-term context with short-term detail, squanders model capacity on resolving incompatible dependencies, and obscures the semantics of the latent dimensions—making directed manipulation or counterfactual sampling nearly impossible.

By contrast, an explicit hierarchical latent structure allocates a dedicated timescale to each layer: a slow-clock top tier encodes global intent or seasonality, intermediate tiers refine meso-level patterns, and a fast-clock bottom tier captures instantaneous noise. This separation prevents semantic mixing, lets the model reuse parameters efficiently, and provides clearly interpretable handles for controllable generation (e.g., editing only high-level intent while preserving fine detail).

## E   Related Works

**Time Series Generation**   Generating realistic time series data [66, 33, 1, 13, 80] is important in numerous fields, including finance, healthcare, and engineering, where access to sufficient data can be challenging. Traditional generative methods, such as Generative Adversarial Networks (GANs) [17, 56, 12, 62, 35] and Variational Autoencoders (VAEs) [39, 46, 7], have been widely explored for time series generation. TimeGAN [75], a GAN-based framework, integrates adversarial and supervised losses to effectively capture temporal dynamics, demonstrating superior performance over previous methods in terms of similarity and predictive performance. However, such GAN-based models often face challenges such as training instability and mode collapse. To overcome these limitations, VAEs have been explored as a more robust alternative. TimeVAE [10] incorporates domain knowledge to model temporal patterns like trends and seasonalities, improving both interpretability and training efficiency. Recent advances have also introduced diffusion-based models [41, 15, 21], such as Diffusion-TS [76], which leverage disentangled temporal representations and Fourier-based training objectives to generate high-quality and interpretable time series while mitigating common limitations of autoregressive approaches. Furthermore, Koopman VAEs (KoVAE) [58] leverage linear latent dynamics inspired by Koopman theory, enabling the integration of domain knowledge and stability analysis. These methods collectively illustrate the evolving landscape of time series generation, with a focus on balancing realism, interpretability, and computational efficiency. Other relevant works include [36], which synthesize time series by manipulating existing time series.

**Identifiability of Causal Representation Learning**   Independent Component Analysis (ICA) has been widely used to identify latent variables with identifiability guarantees, traditionally assuming a linear mixing process [8, 30, 27]. However, this linearity constraint restricts its applicability in more complex scenarios. To handle nonlinear scenarios, researchers have proposed nonlinear ICA by introducing additional assumptions, such as auxiliary variables [37, 72, 47] or structural sparsity [79, 78, 49]. Specifically, methods based on auxiliary variables [28, 29, 50] typically assume

that latent sources are conditionally independent given auxiliary information, such as domain indices or temporal segments, thereby enabling identifiability. For unsupervised approaches, structural sparsity in the generative process has been exploited to recover causal latent factors without relying on auxiliary variables [79, 78, 67]. This approach ensures identifiability by enforcing sparsity constraints on the Jacobian of the mixing function, allowing for the identification up to component-wise transformations and permutations.

Temporal observations introduce unique challenges to identifiability, especially in the presence of instantaneous dependencies and dynamic latent structures. [28] achieves identifiability for stationary data using permutation-based contrastive learning, while [29] extends this approach to nonstationary time series by leveraging variability in data segments. Recent methodss [73, 74] assume conditional independence of latent variables given their time-delayed parent to identify latent causal relations. However, these assumptions often fail in real-world scenarios where instantaneous dependencies are prevalent, such as in human motion data, making them of limited applicability in such contexts. Recently, [48] proposed an identification framework for instantaneous latent dynamics, introducing a sparse influence constraint to enforce sparsity in both time-delayed and instantaneous causal relationships among latent processes. While these advances enhance identifiability, addressing hierarchical latent dependencies remains a key challenge for further improving the identifiability of causal structures in temporal data.

# F    Implementation Details

## F.1    Prior Likelihood Derivation

We first consider the prior of $\ln p(\mathbf{z}_{1:t})$. We start with an illustrative example of stationary latent causal processes with two time-delay latent variables, i.e. $\mathbf{z}_t = [z_{t,1}, z_{t,2}]$ with maximum time lag $L = 1$, i.e., $z_{t,i} = f_i(\mathbf{z}_{t-1}, \epsilon_{t,i})$ with mutually independent noises. Then we write this latent process as a transformation map $\mathbf{f}$ (note that we overload the notation $f$ for transition functions and for the transformation map):

$$
\begin{bmatrix} z_{t-1,1} \\ z_{t-1,2} \\ z_{t,1} \\ z_{t,2} \end{bmatrix} = \mathbf{f} \left( \begin{bmatrix} z_{t-1,1} \\ z_{t-1,2} \\ \epsilon_{t,1} \\ \epsilon_{t,2} \end{bmatrix} \right).
$$

By applying the change of variables formula to the map $\mathbf{f}$, we can evaluate the joint distribution of the latent variables $p(z_{t-1,1}, z_{t-1,2}, z_{t,1}, z_{t,2})$ as

$$
p(z_{t-1,1}, z_{t-1,2}, z_{t,1}, z_{t,2}) = \frac{p(z_{t-1,1}, z_{t-1,2}, \epsilon_{t,1}, \epsilon_{t,2})}{|\det \mathbf{J_f}|}, \tag{A48}
$$

where $\mathbf{J_f}$ is the Jacobian matrix of the map $\mathbf{f}$, where the instantaneous dependencies are assumed to be a low-triangular matrix:

$$
\mathbf{J_f} = \begin{bmatrix} 1 & 0 & 0 & 0 \\ 0 & 1 & 0 & 0 \\ \frac{\partial z_{t,1}}{\partial z_{t-1,1}} & \frac{\partial z_{t,1}}{\partial z_{t-1,2}} & \frac{\partial z_{t,1}}{\partial \epsilon_{t,1}} & 0 \\ \frac{\partial z_{t,2}}{\partial z_{t-1,1}} & \frac{\partial z_{t,2}}{\partial z_{t-1,2}} & \frac{\partial z_{t,2}}{\partial \epsilon_{t,1}} & \frac{\partial z_{t,2}}{\partial \epsilon_{t,2}} \end{bmatrix}.
$$

Given that this Jacobian is triangular, we can efficiently compute its determinant as $\prod_i \frac{\partial z_{t,i}}{\epsilon_{t,i}}$. Furthermore, because the noise terms are mutually independent, and hence $\epsilon_{t,i} \perp \epsilon_{t,j}$ for $j \neq i$ and $\epsilon_t \perp \mathbf{z}_{t-1}$, so we can with the RHS of Equation (A48) as follows

$$
p(z_{t-1,1}, z_{t-1,2}, z_{t,1}, z_{t,2}) = p(z_{t-1,1}, z_{t-1,2}) \times \frac{p(\epsilon_{t,1}, \epsilon_{t,2})}{|\mathbf{J_f}|} = p(z_{t-1,1}, z_{t-1,2}) \times \frac{\prod_i p(\epsilon_{t,i})}{|\mathbf{J_f}|}. \tag{A49}
$$

Finally, we generalize this example and derive the prior likelihood below. Let $\{r_i\}_{i=1,2,3,\cdots}$ be a set of learned inverse transition functions that take the estimated latent causal variables, and output the noise terms, i.e., $\hat{\epsilon}_{t,i} = r_i(\hat{z}_{t,i}, \{\hat{\mathbf{z}}_{t-\tau}\})$. Then we design a transformation $\mathbf{A} \rightarrow \mathbf{B}$ with low-triangular Jacobian as follows:

$$
\underbrace{[\hat{\mathbf{z}}_{t-L}, \cdots, \hat{\mathbf{z}}_{t-1}, \hat{\mathbf{z}}_t]^\top}_{\mathbf{A}} \text{ mapped to } \underbrace{[\hat{\mathbf{z}}_{t-L}, \cdots, \hat{\mathbf{z}}_{t-1}, \hat{\epsilon}_{t,i}]^\top}_{\mathbf{B}}, \text{ with } \mathbf{J_{A \rightarrow B}} = \begin{bmatrix} \mathbb{I}_{n_s \times L} & 0 \\ * & \text{diag}\left(\frac{\partial r_{i,j}}{\partial \hat{z}_{t,j}}\right) \end{bmatrix}.
$$
$$\tag{A50}$$

Similar to Equation (A49), we can obtain the joint distribution of the estimated dynamics subspace as:

$$\log p(\mathbf{A}) = \underbrace{\log p(\hat{\mathbf{z}}_{t-L}, \cdots, \hat{\mathbf{z}}_{t-1}) + \sum_{i=1}^{n_s} \log p(\hat{\epsilon}_{t,i})}_{\text{Because of mutually independent noise assumption}} + \log(|\det(\mathbf{J}_{\mathbf{A}\to\mathbf{B}})|) \quad (A51)$$

Finally, we have:

$$\log p(\hat{\mathbf{z}}_t | \{\hat{\mathbf{z}}_{t-\tau}\}_{\tau=1}^L) = \sum_{i=1}^{n_s} p(\hat{\epsilon}_{t,i}) + \sum_{i=1}^{n_s} \log |\frac{\partial r_i}{\partial \hat{z}_{t,i}}| \quad (A52)$$

Since the prior of $p(\hat{\mathbf{z}}_{t+1:T} | \hat{\mathbf{z}}_{1:t}) = \prod_{i=t+1}^{T} p(\hat{\mathbf{z}}_i | \hat{\mathbf{z}}_{i-1})$ with the assumption of first-order Markov assumption, we can estimate $p(\hat{\mathbf{z}}_{t+1:T} | \hat{\mathbf{z}}_{1:t})$ in a similar way.

## F.2 Evident Lower Bound

In this subsection, we show the evident lower bound. We first factorize the conditional distribution according to the Bayes theorem.

$$\ln p(\mathbf{x}_{1:T}) = \ln \frac{p(\mathbf{x}_{1:T}, \mathbf{z}_{1:T})}{p(\mathbf{z}_{1:T} | \mathbf{x}_{1:T})} = \mathbb{E}_{q(\mathbf{z}_{1:T} | \mathbf{x}_{1:T})} \ln \frac{p(\mathbf{x}_{1:T}, \mathbf{z}_{1:T}) q(\mathbf{z}_{1:T} | \mathbf{x}_{1:T})}{p(\mathbf{z}_{1:T} | \mathbf{x}_{1:T}) q(\mathbf{z}_{1:T} | \mathbf{x}_{1:T})}$$
$$\geq \underbrace{\mathbb{E}_{q(\mathbf{z}_{1:T} | \mathbf{x}_{1:T})} \ln p(\mathbf{x}_{1:T} | \mathbf{z}_{1:T})}_{L_r} - \underbrace{D_{KL}(q(\mathbf{z}_{1:T} | \mathbf{x}_{1:T}) || p(\mathbf{z}_{1:T}))}_{L_{KLD}} = ELBO. \quad (A53)$$

## F.3 Model Details

### F.3.1 Reproducibility of Simulation Experiment

For the implementation of baseline models, we utilized publicly released code for TDRL, CaRiNG. Since the source code of IDOL is not available, we implemented it based on TDRL with the sparsity constraint. And the proposed CHiLD is also modified based on the code of TDRL, and shared hyperparameters remain the same.

### F.3.2 Reproducibility of Real-world Experiment

We follow the setting of [76]. The model details of the proposed method are shown in Table A5. As for other baselines like KoVAE, Diffusion-TS, TimeGAN, and TimeVAE, we employ the official implementations and use the default hyperparameters for a fair comparison. To achieve the results from the well-fit baselines, we employed the default hyperparameters and tried different values of learning rate for the best models. And the number of latent variables on each real-world dataset is shown in Table A6.

# G Experiment Details

## G.1 Simulation Experiment

### G.1.1 Data Generation Process

As for the temporally latent processes, we use MLPs with the activation function of LeakyReLU to model the sparse time-delayed. That is:

$$z_{t,i}^1 = (LeakyReLU(W_{i,:}^1 \cdot \mathbf{z}_{t-1}^1, 0.2) + V_{<i,i} \cdot \mathbf{z}_{t,<i}^2) + \epsilon_{t,i}^1$$
$$z_{t,i}^2 = LeakyReLU(W_{i,:}^2 \cdot \mathbf{z}_{t-1}^2, 0.2) + \epsilon_{t,i}^2, \quad (A54)$$

where $W_{i,:}^*$ is the $i$-th row of $W^*$ and $V_{<i,i}$ is the first $i-1$ columns in the $i$-th row of $V$. Moreover, each independent noise $\epsilon_{t,i}$ is sampled from the distribution of normal distribution. We further let the data generation process from latent variables to observed variables be MLPs with the LeakyReLU units. And the generation procedure can be formulated as follows:

$$\mathbf{x}_t = LeakyReLU((LeakyReLU(\mathbf{z}_t^1, 0.2)) + \epsilon_t^0, 0.2), \quad (A55)$$

Table A5: Architecture details. $T$, length of time series. $|\mathbf{x}_t|$: input dimension. $n$: latent dimension. LeakyReLU: Leaky Rectified Linear Unit. Tanh: Hyperbolic tangent function.

| Configuration | Description | Output |
|---|---|---|
| $\phi$ | Latent Variable Encoder | |
| Input:$\mathbf{x}_{1:t}$ | Observed time series | Batch Size$\times$t$\times$ **x** dimension |
| Convolution neural networks | $|\mathbf{x}_t|$ neurons | Batch Size$\times$t$\times|\mathbf{x}_t|$ |
| Concat zero | concatenation | Batch Size$\times$T$\times|\mathbf{x}_t|$ |
| Dense | n neurons | Batch Size$\times$T$\times$n |
| $\psi$ | Decoder | |
| Input:$\mathbf{z}_{1:T}$ | Latent Variable | Batch Size$\times$T$\times$n |
| Dense | $|\mathbf{x}_t|$ neurons, Tanh | Batch Size$\times$T$\times|\mathbf{x}_t|$ |
| r | Modular Prior Networks | |
| Input: $\mathbf{z}_{1:T}$ | Latent Variable | Batch Size$\times$(n+1) |
| Dense | 128 neurons,LeakyReLU | (n+1)$\times$128 |
| Dense | 128 neurons,LeakyReLU | 128$\times$128 |
| Dense | 128 neurons,LeakyReLU | 128$\times$128 |
| Dense | 1 neuron | Batch Size$\times$1 |
| Jacobian Compute | Compute log(det(J)) | Batch Size |

Table A6: Number of latent variables on each real-world dataset

| | ETTh | Stocks | MuJoco | fMRI | Box | Gestures | Throwcatch | Discussion | Purchases | WalkDog |
|---|---|---|---|---|---|---|---|---|---|---|
| Sample Size | 7 524 | 3 686 | 10 000 | 10 000 | 1 889 | 1 687 | 1 022 | 13 760 | 7 481 | 10 087 |
| Dimension of High-level Layer | 3 | 2 | 4 | 20 | 15 | 15 | 15 | 25 | 25 | 25 |
| Dimension of Low-level Layer | 4 | 4 | 10 | 30 | 30 | 30 | 30 | 26 | 26 | 26 |
| Number of Layer | 2 | 2 | 2 | 2 | 2 | 2 | 2 | 2 | 2 | 2 |

where the dimension of $\epsilon_t^0$ is 1.

We generate six different synthetic datasets. For dataset A, the first layer contains 4 latent variables and the second layer contains 1 latent variable. For dataset B, all the latent variables are in the first layer, since there is only one latent layer. For dataset C, the first layer contains 8 latent variables and the second layer contains 2 latent variables. For dataset D, we consider the second-order latent markov process, and the first and second layers contain 4 and 1 latent variables, respectively. For dataset E, we consider a more complex generation procedure. Specifically, each causal relation and generation using a 2-layer MLP with LeakyReLU activation, which are shown as follows:

$$
\begin{aligned}
z_{t,i}^2 &= M^1(M^2(\mathbf{z}_{t-1}^2)) + \epsilon_{t,i}^2 \\
z_{t,i}^1 &= 0.5 * M^3(M^4(\mathbf{z}_{t-1}^1)) + 0.5 * M^4(M^5(\mathbf{z}_{t-1}^2)) + \epsilon_{t,i}^1 \\
\mathbf{x}_{t,i} &= M^6(M^7(\mathbf{z}_t^1)) + \epsilon_t^0,
\end{aligned}
\tag{A56}
$$

where $M^*$ is a linear layer followed by a LeakyReLU activation. For dataset F, we consider three latent layers with 1, 2, and 4 latent variables. For dataset G, we consider the case with large number of latent layers and dimensions, which contains three latent layers with 8, 8, and 8 latent variables, respectively. The total size of the dataset is 100,000, with 1,024 samples designated as the validation set. The remaining samples are the training set.

### G.1.2  Evaluation Metrics

To evaluate the identifiability performance of our method under instantaneous dependencies, we employ the Mean Correlation Coefficient (MCC) between the ground-truth $\mathbf{z}_t$ and the estimated $\hat{\mathbf{z}}_t$. A higher MCC denotes a better identification performance the model can achieve. In addition, we also draw the estimated latent causal process to validate our method. Since the estimated transition function will be a transformation of the ground truth, we do not compare their exact values, but only the activated entries.

### G.1.3 More Simulation Experiments of Different Length of Observations

To evaluate our theoretical results that $L$-layer latent variables require $2L+1$ consecutive observations, we designed an experiment on a synthetic two-layer model with 8 latent variables in each layer. In our experimental setting, we vary the number of consecutive observations, i.e., 1,3,5, denoted by settings J1, J3, and J5, respectively. The results are shown in Table A7:

Table A7: MCC performance under different numbers of consecutive observations.

|   | Setting | MCC (mean ± std) |
|---|---------|------------------|
| 1 | J1 | $0.6859 \pm 0.024$ |
| 3 | J3 | $0.7001 \pm 0.008$ |
| 5 | J5 | $0.7622 \pm 0.004$ |

According to the experiment results, we can find that we observe that as the length of the consecutive observation window increases, the recovery performance of the latent variables improves. In particular, when the length is $2L + 1$, i.e., 5, the identifiability result is optimal.

### G.2 Real-world Experiment

#### G.2.1 Dataset Description

The detailed descriptions of the datasets are shown as follows:

- Stock is the Google stock price data from 2004 to 2019. Each observation represents one day and has 6 features.

- ETTh1 dataset contains the data collected from electricity transformers, including load and oil temperature that is recorded every 15 minutes between July 2016 and July 2018.

- fMRI is a benchmark for causal discovery, which consists of realistic simulations of blood-oxygen-level-dependent (BOLD) time series.

- MuJoCo (multi-joint dynamics with contact) is a time series dataset generated by the physics engine.

- Huaman 3.6 is collected over 3.6 million different human poses, viewed from 4 different angles, using an accurate human motion capture system. The motions were executed by 11 professional actors, and covered a diverse set of everyday scenarios including conversations, eating, greeting, talking on the phone, posing, sitting, smoking, taking photos, waiting, and walking in various non-typical scenarios. We randomly use four motions, i.e., Gestures (Ge), Jog (J), CatchThrow (CT), and Walking (W).

- HuamnEVA-I comprises 3 subjects, each performing several action categories. Each pose has 15 joints with three axis. We choose 6 motions, i.e., Discussion (D), Greeting (Gr), Purchases (P), SittingDown (SD), Walking (W), and WalkTogether (WT) for the task of human motion forecasting. Specifically, the ground truth motion of the body was captured using a commercial motion capture (MoCap) system from ViconPeak 5 The system uses reflective markers and six 1M-pixel cameras to recover the 3D position of the markers and thereby estimate the 3D articulated pose of the body. We consider the joints as latent variables and the signals recorded from the system as observations.

- Weather [7] dataset offers 10-minute summaries from an automated rooftop station at the Max Planck Institute for Biogeochemistry in Jena, Germany.

- WeatherBench [8] is a benchmark dataset for data-driven medium-range weather forecasting. It repackages forty years (1979-2018) of ERA5 global reanalysis into machine-learning-ready NetCDF tensors sampled every six hours. Fourteen core surface and pressure-level variables—geopotential height, temperature, humidity, wind components, vorticity, potential vorticity, cloud cover, precipitation, solar radiation, and more—are provided on latitude-longitude grids of 0.25°, 1.406°, 2.812°, and 5.625°. Static fields such as land-sea mask, soil type, orography, and grid coordinates are included.

---

[7]https://www.bgc-jena.mpg.de/wetter/
[8]https://agupubs.onlinelibrary.wiley.com/doi/full/10.1029/2020MS002203

- CESM2 [9] delivers 100 fully coupled Earth system simulations at 1° resolution spanning 1850-2100 under CMIP6 historical and SSP3-7.0 forcing. Ensemble spread arises from distinct ocean–atmosphere initial states: ten members seeded from evenly spaced low-drift periods; eighty members created by tiny 10-14 K atmospheric perturbations applied to four AMOC-phased control years; and ten MOAR members useful for regional downscaling.

### G.2.2 Evaluation Metric

- Context-Frechet Inception Distance (Context-FID) score [35] quantifies the quality of the synthetic time series samples by computing the difference between representations of time series that fit into the local context.

- Correlational score [51] uses the absolute error between cross-correlation matrices by real data and synthetic data to assess the temporal dependency.

### G.3 More Experiment Results

### G.3.1 Experiment results on other datasets

We would like to clarify that while some absolute improvements may seem small, e.g., 0.079 (ours) vs. 0.098 (IDOL) in the fmri dataset, it represents a 19% relative improvement. Focusing on percentage improvements, we observe significant gains of 34.07% and 23.91% of two metrics, showing the effectiveness of CHiLD.

Table A8: Experiment results on other datasets.

|  |  | ETTh | Stocks | Mujoco | fmri |
|---|---|---|---|---|---|
| Context-FID Score | CHiLD | 0.111(0.029) | **0.001(0.000)** | 0.238(0.055) | **0.025(0.001)** |
|  | KoVAE | 0.120(0.009) | 0.095(0.013) | 0.024(0.009) | 1.086(0.142) |
|  | Diffusion-TS | 0.116(0.010) | 0.147(0.025) | **0.013(0.001)** | 0.105(0.006) |
|  | TimeGAN | 0.300(0.013) | 0.103(0.013) | 0.563(0.052) | 1.292(0.218) |
|  | IDOL | **0.077(0.013)** | 0.022(0.002) | 0.062(0.012) | 0.065(0.002) |
|  | cwVAE | 0.892(0.059) | 0.807(0.252) | 1.010(0.195) | 0.896(0.103) |
|  | TimeVAE | 0.805(0.186) | 0.215(0.035) | 0.251(0.015) | 14.449(0.969) |
| Correlational Score | CHiLD | 0.008(0.000) | **0.001(0.000)** | **0.001(0.000)** | **0.079(0.003)** |
|  | KoVAE | 0.045(0.006) | 0.007(0.002) | 0.203(0.031) | 11.832(0.007) |
|  | Diffusion-TS | 0.049(0.008) | 0.004(0.001) | 0.193(0.027) | 1.411(0.042) |
|  | TimeGAN | 0.210(0.006) | 0.063(0.005) | 0.886(0.039) | 23.502(0.039) |
|  | IDOL | **0.002(0.000)** | 0.006(0.000) | 0.002(0.000) | 0.098(0.003) |
|  | cwVAE | 0.070(0.001) | 0.053(0.001) | 0.050(0.001) | 18.434(0.030) |
|  | TimeVAE | 0.111(0.020) | 0.095(0.008) | 0.388(0.041) | 17.296(0.526) |

### G.3.2 Ablation Study

To evaluate the effectiveness of each module, we further devise two variants of the proposed CHiLD as follows. 1) **CHiLD-KL**: we remove the $KL$ divergence restriction of prior estimation. 2) **CHiLD-C**: we do not use the context information for the proposed CHiLD. Experiment results are shown in Table A9, we can find that incorporating the contextual observation and KL divergence has a positive impact on the overall performance of the model.

Table A9: Experiment results of different model variants on the Humaneva-Box and Humaneva-Throwcatch datasets.

|  | Box | | Throwcatch | |
|---|---|---|---|---|
| Model | Context-FID Score | Correlational Score | Context-FID Score | Correlational Score |
| CHiLD | **0.007(0.001)** | **0.021(0.002)** | 0.063(0.056) | 0.662(0.007) |
| CHiLD-KL | 0.038(0.003) | 0.637(0.006) | 0.077(0.0044) | 0.974(0.0039) |
| CHiLD-C | 0.017(0.0034) | 0.036(0.0015) | 0.093(0.011) | 1.077(0.0098) |

---

[9]https://climatedata.ibs.re.kr/data/cesm2-lens

# H   Broader Impacts

The proposed **CHiLD** method offers a significant advancement in identifying hierarchical latent dynamics, which is crucial for generating realistic and interpretable time series data across a wide range of domains such as healthcare, finance, climate science, and autonomous systems. By enabling the generation of time series that are both data-driven and causally grounded, our method not only enhances the credibility of the generated sequences but also improves the performance of downstream tasks like forecasting, anomaly detection, and decision-making. These improvements foster increased trust and adoption of machine learning models in critical applications, particularly where reliable data-driven insights are of paramount importance.

Furthermore, our work contributes to the broader field of generative modeling by providing a principled framework for hierarchical latent dynamics. This framework mitigates issues like mode collapse, a common problem in generative models, and enhances the model's generalizability across various datasets. As a result, **CHiLD** stands as a powerful tool for both researchers and practitioners looking to explore complex time series data and apply generative modeling techniques in real-world scenarios. This method paves the way for future research in more complex domains, such as controllable video generation, neuroscience, and predictive analytics in environmental systems, thereby further extending its applicability.

