# OpenReview forum: "Towards Identifiability of Hierarchical Temporal Causal Representation Learning"
_NeurIPS.cc/2025/Conference — NeurIPS 2025 poster_

### Official Review · Reviewer_yDBF · 2025-06-07

**Clarity:** 3
**Significance:** 2
**Originality:** 2
**Rating:** 4
**Confidence:** 4

**Summary:**

This work considers a novel setting that combines hierarchical causal latent structure with temporal dynamics. Under strengthened assumptions on the latent generative model, it proposed a hierarchy of increasingly refined identifiability results based on observations from time steps t-L to t+L, where L is the number of hierarchical layers. The theoretical contributions include i) the block-identifiability of all the variables in time t, ii) block-identifiability of the variables in time t and layer l, and finally iii)element-wise identifiability. To empirically validate the theoretical results, the authors develop a VAE-based model and evaluate it on both synthetic data and nine real-world datasets, including three climate datasets and two human motion datasets. Overall, the proposed method outperforms the other baseline models, including VAE-based methods, generative models, and causal representation learning methods.

**Questions:**

1.	About linear operators, does it exist for arbitrary two distributions? If the answer is no, then assumption (ii) made in Theorem 1, does it implicitly assume that the existence of a linear transformation between the distributions of Lxt+1,··· ,xt+L|zt and Lxt−L,··· ,xt−1|xt+1,··· ,xt+L? The example 4 in appendix B.3, is it a typo? Post-linear in the bold part but non-linear later?

2.	In the hierarchical data generation process, does it assume that the dimensionality of the latent variables at each layer matches that of the observations (since only $n$ is used in the Sec2.1? While from the experimental setting it seems no? Or this is some assumption for theorem only? Does it assume the dimension of latent is known as pior? If no, how to estimate it in the experiment for both real datasets and synthetic?

**Ethical Concerns:**

["NO or VERY MINOR ethics concerns only"]

**Final Justification:**

For completeness

**Limitations:**

Some missing points are listed in the weaknesses above

**Quality:**

3

**Strengths And Weaknesses:**

Strengths

1.	The setting, considering hierarchical causal latent structure with temporal dynamics, is novel and well-motivated. The structure of the series of identifiability results is clear and easy to follow.

2.	Unlike many existing causal representation learning approaches that assume a deterministic mixing process, this work allows for noise in the mixing functions and also cross-layer connections.

3.	The experiments on real datasets are sufficient and well-explained. It matches the setting in the theorem well, showing strong evidence about the efficiency of the effectiveness of the proposed method.

Weaknesses

1.	The theoretical contributions, while clearly presented, offer limited novelty. Both the core assumptions and the proof strategy closely follow prior work.

2.	The assumptions made for achieving identifiability might be restrictive, especially the conditional independence assumption in the final theorem (as the authors also clearly mentioned in the discussion section) and the linear operator at the beginning, which might not be true for many cases.

3.	In the synthetic dataset, the mixing function between x and z is only a simple 2-layer of leakyReLU. I am concerned about whether the setup is sufficiently expressive to convincingly demonstrate the method’s ability to handle complex nonlinear mixing functions.

4.  Although increasing the number of layers in the latent space adds complexity to the identifiability problem, the authors ease this by using a broader temporal window (from t-L to t+L), which introduces a tradeoff and could be impractical in certain applications.

---

> ### Author Rebuttal · Authors · 2025-07-30
>
> Dear Reviewer yDBF, we sincerely appreciate your informative feedback and helpful suggestions that helped clarify our contributions, improve the readability of our theories, and the completeness of our experiments. Please see our point-to-point response below.
>
> >Q1: The theoretical contributions, while clearly presented, offer limited novelty. Both the core assumptions and the proof strategy closely follow prior work.
>
> A1: Thank you for your thoughtful comment, which helps us highlight the contribution of our work. We would like to emphasize that our contribution lies in addressing a fundamentally different and practically motivated problem. Specifically, as illustrated by the motivating example in Appendix C, we aim to address a real-world challenge in representation learning: recovering hierarchical temporal latent structures where only a subset of the latent variables contributes directly to the observations, and where the latent-to-observed mapping is non-invertible and noisy. This setting is not covered by existing results such as those in IDOL, which rely on stronger assumptions (e.g., full invertibility, flat latent structures) and are not applicable in the presence of hierarchical dependencies and partial observability.
>
> To tackle this more realistic yet theoretically challenging problem, we developed a specific model formulation and carefully tailored the proof strategy, introducing additional hierarchical constraints. This allows us to establish identifiability results under broader and more practical conditions. In light of your suggestion, we have highlighted the contribution of our work in the updated version.
>
> >Q2: The assumptions made for achieving identifiability might be restrictive, especially the conditional independence assumption in the final theorem (as the authors also clearly mentioned in the discussion section) and the linear operator at the beginning, which might not be true for many cases.
>
> A2: Thank you for your insightful comment, which helps us further clarify the intuition behind our assumptions and improve the presentation of the paper. First, we would like to clarify that the conditional independence assumption [1,3] and the invertible linear operator [2] assumption are common in the literature of causal representation learning. Below, we elaborate on why these assumptions are not overly restrictive and how they can be further relaxed:
>
> 1. As for the conditional independence assumption, we assume conditional independence within each layer of latent variables, while allowing instantaneous effects across layers. This assumption is primarily made for clarity in presenting the identifiability of latent variables within each layer. Even the the conditional independence assumption does not hold in practice, the latent variables from each layer can still achieve the block-wise identifiability according to Theorem 2, facilitating the hierarchically controllable generation. Moreover, this assumption can be relaxed by leveraging the results of [4], identifiability can still be established by exploiting the sparsity of latent dynamics even in the presence of instantaneous effects.
>
> 2. As for the linear operator assumption, I guess you mean that the linear operator assumption denotes a type of linear transformation. However, the linear operator describes a mapping between any two arbitrary distributions. And the assumption of an injective linear operator, e.g., $L_{x_{t+1},\cdots,x_{t+L}|z_t}$, intuitively means that the information contained in the observations is equivalent to that in the latent variables. They are commonly assumed when we try to recover the joint distribution in the presence of latent variables. Furthermore, it is also a natural assumption for real-world scenarios, since if the temporal observations $x_{1:T}$ do not encode the information of $z_t$, the latent variables cannot be recovered from the observations. In this paper, we explicitly state this assumption to clarify the sufficient conditions under which hierarchical latent variables are identifiable. Additionally, we have provided several **practical examples of injective linear operators in Appendix B.3** to support the plausibility of this assumption.
>
> [1] Hyvarinen, Aapo, and Hiroshi Morioka. "Unsupervised feature extraction by time-contrastive learning and nonlinear ica." NeurIPS2016
> [2] Fu, Minghao, et al. "Identification of Nonparametric Dynamic Causal Structure and Latent Process in Climate System.
> [3] Yao, Weiran, Guangyi Chen, and Kun Zhang. Temporally disentangled representation learning.
> [4] Li, et al. "On the identification of temporally causal representation with instantaneous dependence." (ICLR2025)
>
> >Q3: In the synthetic dataset, the mixing function between x and z is only a simple 2-layer of leakyReLU. I am concerned about whether the setup is sufficiently expressive to convincingly demonstrate the method’s ability to handle complex nonlinear mixing functions.
>
> A3: Thank you for this constructive comment, which improves the soundness of our experiment results. In light of your suggestions, we have provided additional synthetic experiments with different nonlinear mixing functions, including 3- and 4-layer MLPs with Tanh and leakyReLU activation, respectively. Experiment results are shown as follows.
>
> |Mixing Function| MCC |
> |---|---|
> | 3-layer Leakly_ReLU | 0.8024 (0.0291) |
> | 4-layer Leakly_ReLU | 0.7564 (0.0123) |
> | 3-layer Tanh | 0.8118 (0.0217) |
>
>
> >Q4: Although increasing the number of layers in the latent space adds complexity to the identifiability problem, the authors ease this by using a broader temporal window (from t-L to t+L), which introduces a tradeoff and could be impractical in certain applications.
>
>  A4: Thank you for this insightful comment.  We agree that in certain practical applications, obtaining a sufficiently long temporal window may be challenging. This naturally leads to a trade-off between the length of the observation window and the extra assumption required for identifiability.
>
> To illustrate this, let us consider a simplified case with only two layers of latent variables. Instead of requiring 5 consecutive observations as in our main setting, the latent variables can be identified with only 3 time steps of observations under the sparsity assumption. When each latent variable has a sufficient number of pure children (e.g., 2 conditional independent observed variables), we can first identify the lowest-layer latent variables by leveraging the results in [5]. Then, based on the recovered lower-layer representations, the higher-layer latent variables can be identified iteratively. This significantly relaxes the requirement on the temporal window length.
>
> [5] Kong, et al. "Identification of nonlinear latent hierarchical models." NeurIPS2023
>
> >Q5: About linear operators, does it exist for arbitrary two distributions? If the answer is no, then assumption (ii) made in Theorem 1 does it implicitly assume that the existence of a linear transformation between the distributions of Lxt+1,···,xt+L|zt and Lxt−L,···,xt−1|xt+1,···,xt+L? Example 4 in Appendix B.3, is it a typo? Post-linear in the bold part, but non-linear later?
>
> A5: Thanks for your good question. Let us answer it point by point. First, the linear operators exist for arbitrary two distributions as mentioned in [6]. We also thank you for pointing out the typos; the "Post-linear" should be "Post-nonlinear".
>
> [6] Nelson Dunford and Jacob T. Schwartz. Linear Operators. John Wiley & Sons, New York, 1971
>
> >Q6: In the hierarchical data generation process, does it assume that the dimensionality of the latent variables at each layer matches that of the observations (since only n is used in the Sec2.1? While from the experimental setting it seems no? Or is this some assumption for the theorem only? Does it assume the dimension of the latent is known a priori? If no, how to estimate it in the experiment for both real datasets and synthetic?
>
> A6: Thank you for raising this important question, which helps us clarify the assumptions on the dimensionality of latent variables.
>
> First, our theoretical results do not assume that the dimensionality of the latent variables at each layer matches that of the observations. In our experiments, we chose equal dimensionality primarily for simplicity, as it allows us to clearly examine how the relative size of the latent space and observation space affects identifiability. As stated in Lines 1094–1101, the mapping from latent to observed variables is injective as long as the total dimensionality of the observed variables in $x_{t-\tau},\cdots,x_{t-1}$ is greater than or equal to the total dimensionality of the latent variables. This result has also been empirically supported by our synthetic experiments. Following your suggestion, we have added a corollary in the revised version to explicitly state that the latent dimensionality is not restricted in theory and can vary in practice.
>
> Second, we assume the dimensionality of latent variables to be known in our theoretical analysis. For real-world scenarios where the true dimensionality is unknown, we treat the number of latent variables and the number of layers as hyperparameters. In practice, these can be estimated via standard model selection techniques, such as evaluating predictive accuracy or log-likelihood under different configurations, to select the optimal values.

---

> > ### Author Response · Authors · 2025-08-06
> >
> > Dear Reviewer yDBF,
> >
> > Thank you for your valuable time in reviewing our submission and for your insightful suggestions to make our contributions clearer. We've tried our best to conduct the experiments and address your concerns in the response and updated submission. Due to the limited rebuttal discussion period, we eagerly await any feedback you may have regarding these changes. If you have further comments, please kindly let us know--we hope for the possible opportunity to respond to them.
> >
> > Many thanks,
> >
> > Authors of submission #8292

---

> > > ### Comment · Area_Chair_rRAz · 2025-08-07
> > >
> > > Hi,
> > >
> > > The reviewers have not responded because I accidentally briefly revealed some identifying information about two of the reviewers during this rebuttal period. This was a mistake was mine alone (I clicked the wrong button), and not the fault of the reviewers. To avoid any potential negative repercussions for them, the PCs have decided that these reviewers will not take part in the discussion period (so they are not ignoring your comment) and I will also do a complete review of the paper myself and provide it to the SACs with my recommendation to inform their final decision on the paper.
> > >
> > > I am very sorry about the fact that this has result in limited discussion. I will take this all into account when making my decision so the lack of discussion will not negatively impact your paper's chance of acceptance.
> > >
> > > -Your AC

---

> > > > ### Author Response · Authors · 2025-08-07
> > > >
> > > > Dear Area Chair rRAz,
> > > >
> > > > Many thanks for letting us know--we’re now no longer concerned about the two reviewers.  We completely understand the situation and hope the process goes well.
> > > >
> > > > Thank you for your contribution to the community,
> > > >
> > > > Authors of Submission #8292

---

### Official Review · Reviewer_nGAz · 2025-06-26

**Clarity:** 2
**Significance:** 2
**Originality:** 2
**Rating:** 3
**Confidence:** 4

**Summary:**

# Summary

The paper establishes identifiability results for hierarchical temporal causal representations. It presents two main theorems and one lemma:

1. **Theorem 1** : Proves block-wise identifiability of the latent vector $z_t$.
2. **Theorem 2** : Extends block-wise identifiability to each hierarchical layer.
3. **Lemma 1** : Shows component-wise identifiability of individual latent variables $z^l_{i,t}$.

Additionally, the authors propose an algorithm to reconstruct the multilayer latent variables.

**Questions:**

- I remain uncertain about the generative process chosen for the model. Could the authors clarify its motivation? Specifically, what underpins the assumptions that (i) variables in layer $l+2$ cannot influence those in layer $l$ or layer 1, (ii) only a single temporal lag is considered, and (iii) neither $z^{l-1}_{t-1}$ nor $z^{l-2}_t$ can be causal for $z^{l}_t$?

- Several modelling assumptions appear tailored to meet the identifiability conditions rather than arising from domain considerations. Could the authors explain how these assumptions are justified independently of the identifiability proofs, in other words are there any clear real world application of such structure?

- The framework assumes that each observation $x_t$ is generated solely by the first latent layer. Why are contributions from higher layers ruled out, and how realistic is this restriction in practical settings?

- Every latent layer is specified to have dimensionality $n$, matching that of the observed data. What motivates this choice, and how sensitive are the identifiability results to relaxing the dimensionality constraint?

**Ethical Concerns:**

["NO or VERY MINOR ethics concerns only"]

**Final Justification:**

.

**Limitations:**

The authors mentioned one limitation related to their assumptions like layer wise conditional independence. However, as we mentioned, we still have concerns regarding the hierarchical structure assumed by this paper as well as the identifiability techniques used to prove the results.

**Paper Formatting Concerns:**

Nothing to note.

**Quality:**

2

**Strengths And Weaknesses:**

# Strengths

- **Novel hierarchical generative process:** The paper introduces a new multi-layer generative model in which each layer injects additional noise, making the identifiability problem notably more challenging.

- **Methodological contribution:** It thoughtfully adapts techniques from the measurement-error literature to causal latent-variable identifiability.

- **Clear intuition for assumptions:** For every theorem, the authors provide intuition that clarifies each assumption, helping readers in understanding the results.

# Weaknesses and Questions:

- **Clarity and motivation:** The introduction does not yet offer a clear motivation for the specific hierarchical structure used in the paper or its real-world relevance. Concrete examples or use cases would greatly improve the writing and the storytelling of this paper.

- **Originality of exposition:** Portions of the introduction closely resemble text from the IDOL paper, though paraphrased. Clearer attribution or more distinctive framing would enhance transparency and underscore the paper’s unique contributions.

- **Premature technical detail:** Technical discussion of the single-layer assumption appears on the first page without sufficient context. Deferring such material until after the problem statement and motivating examples would aid readability.

- **Generative process assumptions:** Some modeling choices, not allowing influences from layer \(l+2\) to \(l\), limiting interactions to a single lag, and excluding links such as $z^{l-1}_{t-1}$, $z^{l}_{t}$ seem unrealistic. Several assumptions feel tailored to produce identifiability rather than arising from domain knowledge. Stronger justification would help substantiate these assumptions.

- **Observation model:** The framework assumes that only the first latent layer generates \(x_t\). Explaining why higher layers do not contribute, or exploring a more flexible observation mechanism, would improve realism.

- **Layer dimensionality:** Each latent layer is set to have the same dimension \(n\) as the observations. Discussing scenarios where this design choice is appropriate and motivate it

- **Overlap with prior proofs:**   *Theorem 1* closely follows Theorem 1 in “Instrumental Variable Treatment of Nonclassical Measurement Error Models.”  *Theorem 2* adapts the proof strategy of Theorem 1 in the IDOL paper with only minor modifications.

- **Minor errors:** There are typographical issues in Eq(3), Appendix Eq(A19), and the sentence around line 1076 that should be corrected.
- Please correct me if I’m mistaken, identifiability results in CRL identify latent variables $z_t$ and try to prove the existence of an invertible mapping between $\hat{z}_t$ and $z_t$. Given that the observation $x_t$ are observed and fixed can you explain why $\hat{x}_t$ in your proofs ? Using a different symbol for the estimated model Eq(A16) might make the notation clearer.

- **Operator definition (EqA8):** It would be more rigorous to define the operator $L_{x_{>t}\mid z_t}$ explicitly, as done in *Instrumental Variable Treatment of Nonclassical Measurement Error Models* proof in page 17.

- **Fairness of the IDOL comparison:** The manuscript claims to achieve identifiability without a permutation step, unlike IDOL. However, IDOL’s component-wise identifiability also holds up to permutation, while the present work avoids only layer-index permutation due to its hierarchical design.

---

> ### Author Rebuttal · Authors · 2025-07-30
>
> Dear Reviewer nGAz, we highly appreciate the valuable comments and helpful suggestions on our paper and the time dedicated to reviewing it. Below, please see our point-to-point responses.
>
> >Q1: "Clarity and motivation"
>
> A1: Thank you for your comment. We actually put a lot of effort into the motivation and the clarity, as you see **Page 1, Lines 34–35 in introduction**. Moreover, we have provided motivations and real-world examples in **Line 1243-1268 in Appendix C**. Please kindly let us know if you cannot find it. For information below the summary of the two real-world examples.
> 1. Human Motion Modeling: The higher-level latent variables correspond to coarse movement categories (e.g., walking, running), while the lower-level latent variables capture fine-grained kinematics.
> 2. Climate Data: We consider a three-layer hierarchical structure where the latent variables represent seasonal, monthly, and daily factors that influence weather patterns.
>
> >Q2: "Originality of exposition"
>
> A2: These two papers are in the same line of research, aiming to identify the latent variables of time series data. And therefore, we agree that to some level, the motivations are related. Moreover, we would like to respectfully highlight that the clear difference between these two works is as follows:
> 1. **Different research focus**. IDOL is motivated by identifying latent causal processes with instantaneous dependencies under sparse causal influences. CHiLD specifically targets the identifiability of hierarchical latent dynamics, which has not been addressed previously. The introduction directly starts from hierarchical temporal structures and discusses why existing single-layer methods fail in this setting.
> 2. **Distinct motivation and framing**. IDOL emphasizes relaxing strong assumptions, such as interventions or grouping of observations, to achieve identifiability in the single-layer case. In contrast, we highlight that the commonly used “sufficient changes” condition breaks down under hierarchical structures, thereby motivating the need for a new framework to recover the joint distribution of latent variables.
> 3. **Independent theoretical contributions**. IDOL establishes component-wise identifiability based on a sparse latent process. While CHiLD proposes another block-wise identifiability, showing that the joint distribution of latent variables can be uniquely determined with certain adjacent observations, and further achieves component-wise identifiability within each layer. This theoretical insight is entirely different.
> 4. **Clearer attribution**. While the two works share a general background on causal representation learning, we will revise the introduction to explicitly acknowledge IDOL as a single-layer baseline and further emphasize how CHiLD goes beyond it by addressing hierarchical structures.
>
> In light of your comment, we have also included a clear distinction in the revised version.
>
> >Q3: "Premature technical detail"
>
> A3: We kindly remind you that the technical discussion has been provided in **Section 2.2 (Page 3)** of the original submission with the title of “Why previous theoretical results can hardly identify the hierarchical latent variables?”. We believe this placement aligns with your suggestion to defer technical content until after sufficient context is provided.
>
> >Q4&11&12: As for "Generative process assumptions".
>
> A4: Thank you for noticing this. Kindly let us explain why the model is actually flexible as follows:
>
> 1. Not allowing influences from layer l+2 to l. Although the generative process in Fig. 1 illustrates a first-order Markov structure, we explicitly state on **Line 69 and in the footnote on Page 2** that “the illustration shows a first-order Markov structure; however, our method can extend to higher-order dependencies.” Moreover, as shown in Equ.(2), $z_{t,i}^l=f_i(Pa_e(z_{t,i}^l),Pa_d(z_{t,i}^l), \epsilon_{t,i}^l)$, we does not imply the first-order dependencies, and cross-layer interactions e.g., from layer (l+2) to (l), can be naturally incorporated. Furthermore, when influences from layer (l+2) to (l) exist, we can consider layers (l) to (l+2) as a merged layer, which is a special case of the 1-order Markov structure.
> 2. Limiting interactions to a single lag. As our model is not limited to a first-order Markov structure, we have considered scenarios with multiple temporal lags, and the corresponding proof has been provided in **Appendix B.6**.
> 3. Excluding links such as $z_{t-1}^{l-1} \to z_t^l$. If there is a link from $z_{t-1}^{l-1}$, we can only achieve the block-wise identifiability results, where $z_{t-1}^{l-1}$ and $z_t^l$ compose a block. However, we should make such an assumption to establish very strong component-wise identifiability results. If that's not the case, the result is still non-trivial. However, it is not as strong as the present one. In light of your reminder, we have also added a simulation experiment where there are links from $z_{t-1}^{l-1}$ to $z_t^{l}$, the experiment results are shown as follows:
>
> ||MCC|
> |-|-|
> |w. $z_{t-1}^{t-1}\to z_t^l$ |0.811(0.021)|
> |w.o $z_{t-1}^{t-1}\to z_t^l$ |0.631(0.019)|
>
> Experiment results show that the component-wise identification is hard to achieve when there is $z_{t-1}^{l-1}\to z_t^l$. To summarize, strong identifiability results rely on appropriate assumptions. However, we have discussed the consequences of the violation of this particular assumption in the revised version.
>
> In light of your reminder, we have highlighted the aforementioned cases in the updated version.
>
> >Q5&13: "Observation model"
>
> A5: Thank you for this valuable comment. We would like to clarify that the assumption that only the first latent layer directly generates the observations $x_t$ is motivated by hierarchical latent processes often encountered in real-world scenarios, such as climate data. In such settings, high-level latent variables govern abstract temporal dynamics, which are manifest in the observed data via the bottom latent variables.
> Moreover, this case is more challenging, as some latent variables do not directly contribute to $x_t$. Motivated by the real problems, this constraint allows us to establish the identifiability of the joint distribution, even in the presence of noise, because the latent variable cannot be recovered as determining functions.
>
> >Q6&14: "Layer dimensionality"
>
> Q6: Thanks! First, our theoretical results do not require the dimensionality of the latent variables at each layer to match that of the observations. The same dimensionality was chosen in our work primarily for convenience, as it allows us to explicitly analyze how the relationship between latent dimensionality and observed dimensionality affects identifiability. As stated in Lines 1094–1101, the injectivity is ensured when the total observed dimensionality in $x_{t-\tau},\cdots,x_{t-1}$ is greater than or equal to the total latent dimensionality, which is also been verified in our synthetic experiments. We have added a further discussion in the revised version to explicitly emphasize that the latent dimensionality is flexible in practice.
>
> >Q7: "Overlap with prior proofs"
>
> A7: Thank you very much! As mentioned above, we aim to address a real problem in machine learning as you can see the motivation example in Appendix C. And this problem has not been handled in previous works like IDOL. These prior results do not account for the hierarchical temporal structure or the form of non-invertibility and noisy mixing structure in our setting.
>
> To achieve identifiability in new cases, we have to formulate the specific model and then tailor the proof and incorporate further constraints to achieve our identifiability results. We hope you will find this work serves as an essential step in dealing with important problems in representation learning.
>
> >Q8: "Minor errors"
>
> A8: Thanks! We would like to clarify the following:
> 1. Eq(3): There is no typographical error here. Eq(3) is correct as it explicitly expresses that the top-layer latent variables are only influenced by time-delayed variables, consistent with the generative process assumptions.
> 2. Appendix Equation (A19): You are correct, and we have revised it in the updated version to $\int_ {\hat{X}_ t}p_ {\hat{x}_ t|\hat{z}_ t}d\hat{x}_ t$.
> 3. Sentence around Line 1076. We guess you are confused with $D_ {\hat{x}_ t|\hat{z}_ t}$ 和 $D_ {x_ t|z_ t}$, but they are correct since they represent the estimated conditional distribution and the true conditional distribution, respectively.
>
> >Q9: The explaination of $\hat{x}$.
>
> A9: Thank you for your question. We use the “^” notation to denote estimated variables, which is a convention widely adopted in related literature[1]. Specifically, $\hat{x}_t$ in our proofs refers to the estimated observed variables from the learned generative model. In light of your suggestion, we have added a clarification in the notation table of the updated version to explicitly state this point.
>
> [1] Score-based Causal Representation Learning: Linear and General Transformations
>
> >Q10: Operator definition (EqA8)...
>
> A10: Thank you for your helpful suggestion. In the original version, we expressed the linear operator in terms of its probabilistic form for better intuitive understanding. We have now explicitly defined the operator $L_{x>t|z_t}$ in the updated version.
>
> >Q11: "Fairness of the IDOL comparison"
>
> A11: Thank you for pointing out the connection. Our results without layer-wise permutation benefit from the hierarchical prior. This prior reflects the natural organization of many real-world temporal processes and allows us to distinguish layers without permutation. Furthermore, compared with IDOL, our model allows for a noisy mixing procedure where the latent variables are not deterministic functions of the observations, and some latent variables may not directly contribute to $x_t$. This setting is more aligned with many real-world problems, where the generation process involves uncertainty.

---

> > ### Author Response · Authors · 2025-08-06
> >
> > Dear reviewer nGAz,
> >
> > Thanks for the time you dedicated to carefully reviewing this paper. It would be highly appreciated if you let us know whether our responses properly address your concerns, despite your busy schedule. Thanks a lot!
> >
> > Best regards,
> >
> > Authors of submission 8292

---

### Official Review · Reviewer_9SP8 · 2025-07-02

**Clarity:** 2
**Significance:** 3
**Originality:** 4
**Rating:** 5
**Confidence:** 1

**Summary:**

This submission addresses the problem of performing inference in a hierarchical latent causal process given time series data. The authors establish the component-wise identifiability of the latent variables under such a model, and develop a variational autoencoder architecture that produces the correct conditional independencies in the generation model and can accurately recover the latent posterior at inference time. The resulting generative model achieves state-of-the-art performance, while utilizing the hierarchical structure to disentangle levels of abstraction at generation time. This separation allows for controllable generation.

**Questions:**

Does performance degrade when the number of layers is increased above 2?

**Ethical Concerns:**

["NO or VERY MINOR ethics concerns only"]

**Final Justification:**

See discussion. The added experiment further demonstrates the benefits of their model's hierarchical structure. I am keeping my score at 5 and believe this submission should be accepted.

**Limitations:**

yes

**Quality:**

4

**Strengths And Weaknesses:**

Strengths:

- The paper provides extensive theoretical contributions that establish the identifiability of latent variables under a hierarchical causal process.

- The experiments are promising, and show widespread state-of-the-art performance on simulated and real data.

Weaknesses:

1. I found some parts of the submission to lack clarity, although I acknowledge that these issues might be a result of a lack of confidence in this specific setting. For example, Definitions 1 & 2 seem to be missing key conditions: $\hat{z}_t^l$ needs to be defined as the estimated latent variable from observations, or else that statement "there exists an invertible function $h$ and value $\hat{z}$ such that $z=f(\hat{z})$" is trivially true.

2. On real world data, only $L=2$ is explored, reducing the impact of working with a hierarchical latent process.

---

> ### Author Rebuttal · Authors · 2025-07-30
>
> Dear Reviewer 9SP8, we are very grateful for your valuable comments, helpful suggestions, and encouraging feedback. They have greatly helped us improve the readability of our paper and enhance the completeness of our experiments. As per your valuable suggestions, we have provided a clearer definition and added more real-world experiments to better evaluate the different layers of latent variables.
>
> >Q1: I found some parts of the submission to lack clarity, although I acknowledge that these issues might be a result of a lack of confidence in this specific setting. For example, Definitions 1 & 2 seem to be missing key conditions: $\hat{z}_t^1$ needs to be defined as the estimated latent variable from observations, or else that statement "there exists an invertible function $h$ and value $\hat{z}_t$ such that $z_t=f(\hat{z}_t)$" is trivially true.
>
> A1: Thank you for pointing this out. In light of the suggestion, we have revised Definitions 1 and 2 to explicitly state that $\hat{z}_t$ denotes the estimated latent variables inferred from observations.
>
> >Q2: On real-world data, only $L=2$ is explored, reducing the impact of working with a hierarchical latent process.
>
> A2: Thank you for this insightful comment. In light of your suggestions, we have conducted additional experiments with 3 and 4 latent layers on five real-world datasets: Weather, WeatherBench, Throwcatch, Gestures, and CESM2. Experiment results are shown as follows.
>
> | Dataset | Weather | WeatherBench | Throwcatch | Gestures | CESM2 |
> |:---:|:---:|:---:|:---:|:---:|:---:|
> | # of Levels | Context-FID |  |  |  |  |
> | 1 | 0.676(0.103) | 0.088(0.014) | 0.025(0.004) | 0.041(0.006) | 0.126(0.023) |
> | 2 | 0.507(0.042) |  0.078(0.017) |  0.022(0.003) | 0.032(0.004) | 0.018(0.003) |
> | 3 | 0.486(0.199) | 0.068(0.013) | 0.021(0.002) | 0.026(0.005) | 0.034(0.004) |
> | 4 | 0.270(0.035) | 0.063(0.019) | 0.029(0.005) | 0.044(0.007) | 0.042(0.010) |
> | # of Levels | Correlational Score |  |  |  |  |
> | 1 | 0.169(0.003) | 5.476(0.025) | 0.093(0.009) | 0.112(0.005) | 1.520(0.025) |
> | 2 | 0.165(0.004) |  5.190(0.016) | 0.065(0.001) |  0.089(0.003) | 1.250(0.009)  |
> | 3 | 0.155(0.007) | 4.952(0.017) | 0.080(0.003) | 0.082(0.004) | 1.521(0.023) |
> | 4 | 0.158(0.005) | 4.770(0.015) | 0.082(0.002) | 0.095(0.009) | 1.638(0.049) |

---

> > ### Comment · Reviewer_9SP8 · 2025-08-02
> >
> > Thank you for your response and for the added experiments. The new table adds context as to the effect of increasing hierarchical depth. I will keep my score at 5.

---

> > > ### Author Response · Authors · 2025-08-03
> > > **Official Comment by Authors**
> > >
> > > We are very happy that you found the response well addressed your concerns. Thank you once again for your valuable comments and suggestions and for championing our submission.
> > >
> > > With best wishes,
> > >
> > > Authors of submission #8292

---

### Official Review · Reviewer_nYtx · 2025-07-05

**Clarity:** 3
**Significance:** 2
**Originality:** 3
**Rating:** 5
**Confidence:** 2

**Summary:**

This paper investigates the problem of hiearchical temporal CRL, which investigates a dynamic latent Bayesian network where the observations are generated by the last layer of latent variables, which itself is generated by its 1-level higher and 1-step previous blocks of latent variables. They show that under regularity conditions, identifying the entire latent space with $L$ layers is possible using the joint distribution of $2L+1$ consecutive observation values. They then apply existing ideas to further refine this result to per-block and per-component recovery.

**Questions:**

- Even if it is obvious, can the authors include a formal equivalence, consistency or "closeness" statement for the implementation method?
- In the spirit of Appendix G.1.3, can you show that adding more than necessary information does not significantly increase or reduce performance?
- Please add a clarification in experiments section about (i) the synthetic experiments showing ID performance and (ii) real-world experiments showing data synthesis performance; this knowledge shouldn't be expected from a reader but written explicitly. I suggest slightly renaming the subsections to reflect this difference.
- I did not find the compute resources necessary to run the experiments. Can you add it to the appendix in writing? Even if it is negligible -- I know it is not -- simply say so.

**Ethical Concerns:**

["NO or VERY MINOR ethics concerns only"]

**Final Justification:**

The authors have satisfactorily answered all my concerns in the rebuttal/discussion phases. I believe the paper is solid except for motivation/setting. However, I think that the extensive work done in other sections (e.g. experiments) strengthens the motivation enough for acceptance.

**Limitations:**

yes

**Quality:**

3

**Strengths And Weaknesses:**

The main departure of this work from existing temporal CRL papers is the observation model: The observed variables is NOT generated by an invertible transformation of the entire latent space, but, only the final layer of it. As far as I am aware, this way of inducing a lack of invertibility -- not due to a measurement noise, but from actual missing information in latent space -- is new, and perhaps promising in the context of real-world applications.

While I appreciate the extensive study done for the experimentation, I find the overall workflow of converting identifiability results to theory-inspired VAE implementations slightly backward. Even if it is obvious, including a formal equivalence, consistency or "closeness" statement for the implementation method the authors describe would make the contribution -- and the overall flow of the paper -- more concrete.

---

> ### Author Rebuttal · Authors · 2025-07-30
>
> Dear Reviewer nYtx, we sincerely appreciate your encouraging and insightful feedback, which has significantly helped us enhance the completeness of our theoretical results and soundness of our experimental results. We have added a consistency result for the estimation and additional experimental results with different numbers of consecutive observations. Please find our point-by-point responses below.
>
> >Q1&Q2 (include a formal equivalence, consistency, or "closeness" statement for the implementation method): While I appreciate the extensive study done for the experimentation, I find the overall workflow of converting identifiability results to theory-inspired VAE implementations slightly backward. Even if it is obvious, including a formal equivalence, consistency, or "closeness" statement for the implementation method the authors describe would make the contribution -- and the overall flow of the paper -- more concrete.
>
>
> A1 and A2: Thank you for your thoughtful feedback. In light of your suggestion, we have provided a consistency theoretical result for our implementation in the updated version, which is shown as follows:
>
> Theorem (**Consistency of Estimation**) Suppose the following assumptions hold:
> 1) The variational posterior distribution $q(z_t|x_{t-L:t+L})$ is expressive enough to contain the prior $p(z_t|z_{t-1:t-\tau})$.
> 2) The ELBO is optimized jointly with respect to both the generative parameters and inference parameters.
>
> Then, in the limit of infinite data, the estimated latent variables are identified up to permutation and component-wise invertible transformations.
>
> Proof: The ELBO can be written as follows:
>
> $ELBO=\mathbb{E}_ {q(z_ {1:T}|x_ {1:T})}\sum_ {t=1}^T\log p(x_ t|z_ t) + \mathbb{E}_ {q(z_{1:T}|x_{1:T})}[\sum_ {t=1}^T(\log p(z_ t|z_ {t-1:t-\tau}))-\log q(z_ t|x_ {t-L:t+L})]$
>
> Under assumption (1), the true conditional prior lies within the variational family, so the KL divergence term can be minimized to zero as the model capacity and data size grow. In this case, optimizing the ELBO is equivalent to maximizing the log-likelihood, and the VAE inherits the consistency properties of maximum likelihood estimation. Moreover, since our identifiability results (Theorems 1–2) guarantee that the latent variables are identifiable up to an invertible transformation, the estimated latent variables converge to an equivalent representation of the true latent variables in the limit of infinite data.
>
>
> >Q3: In the spirit of Appendix G.1.3, can you show that adding more than necessary information does not significantly increase or reduce performance?
>
> A3: Thank you for this insightful suggestion, which helps strengthen the robustness of our experimental validation. To assess the impact of including more than the necessary amount of information, we conducted additional experiments on a synthetic dataset with three-layer hierarchical latent dynamics. Specifically, we varied the number of consecutive observations: 1, 3, 5, 7, and 9 time steps. The corresponding results in terms of MCC are reported below:
>
> | Length of Observations | 1 | 3 | 5 | 7 | 9 |
> |---|---:|---:|---:|---:|---:|
> | MCC | 0.6859(0.024) | 0.7001(0.008) | 0.8042(0.004) | 0.8241(0.015) | 0.8266(0.023)|
>
> This indicates that once a sufficient amount of information is available to recover the latent structure, adding more data does not significantly affect the identification performance, which aligns with the intuition that excessive redundancy contributes little additional benefit.
>
> >Q4: Please add a clarification in the experiments section about (i) the synthetic experiments showing ID performance and (ii) real-world experiments showing data synthesis performance; this knowledge shouldn't be expected from a reader but written explicitly. I suggest slightly renaming the subsections to reflect this difference.
>
> A4: Thank you so much for your insights! In light of your suggestions, we have explicitly added a clarification in Section 5. Specifically, the synthetic experiments are designed to evaluate the identifiability performance. i.e., whether the learned latent variables align with the ground-truth latent variables. And the real-world experiments focus on data synthesis performance, demonstrating that the learned latent representations can generate realistic and consistent time series data.
>
> Moreover, we have renamed the subsection titles to better reflect these objectives:
>
> “Experiments on Synthetic Data” → “Synthetic Experiments: Evaluation of Identifiability"
>
> “Experiments on Real-World Data” → “Real-World Experiments: Evaluation of Time Series Generation”
>
> >Q5: I did not find the computing resources necessary to run the experiments. Can you add it to the appendix in writing? Even if it is negligible -- I know it is not -- simply say so.
>
> A5: Thanks a lot! We have provided the compute resources in the updated version. Specifically, we conduct all the experiments on a single NVIDIA GTX 3090 24GB GPU and an i7-7700k CPU with 64Gb memory.

---

> > ### Comment · Reviewer_nYtx · 2025-07-31
> >
> > I thank the authors for their explanations. I am very satisfied with the answers to Q3-5. However, I am not sure about the proof approach provided as response to the first point (Q1-2) for two reasons (I may be wrong on both fronts, please explain why if so):
> > 1. Had minimizing this loss been sufficient for identifiability, you would not require most of your other assumptions (e.g. on the variability of log density derivatives)
> > 2. For the proof itself, the KL term would not vanish due to lack of invertibility induced by $\epsilon^0_t$  (and even perhaps the multi-layer structure). There could be an argument for why this is still fine, but the proof is not there yet.
> >
> > That said, I think lack of such a guarantee is **NOT** a major weakness. I do not like it, but the existing theoretical contributions and experiments are sufficient in my view. Therefore, I retain my rating at 5 - accept.

---

> ### Author Response · Authors · 2025-08-03
> **Thanks for your feedback**
>
> Dear Reviewer nYtx, we are delighted that you see the existing theoretical contributions and experiments are sufficient. Besideds, we are also happy that you are satisfied with the answers to Q3-5. Regarding your further questions, we have provided further explaination as follows:
>
> > Q1: Had minimizing this loss been sufficient for identifiability, you would not require most of your other assumptions (e.g. on the variability of log density derivatives)
>
> A1:Thank you for pointing this out. The consistency theorem actually rely on the assumptions on the true data generating process, including sufficiency change. We have included it explicitly as:
>
> - The data generation process follows data conditions in Theorem 1,2
>
> Moreover, we explain how to incorporate our implementation with the assumptions in Theorem 1 and 2 as follows:
>
> 1. **Bounded, continuous, and positive density assumption:** These assumptions are generally met in practice through the use of neural networks.
>
> 2. **Injective linear operator assumption:** To approximate the injectivity of linear operators, we employ a contextual encoder that maps observations to latent variables, and a decoder that further maps latent variables to observations. This model architecture implicitly models the distributions such as $p(z_t|x_{t+1},\cdots,x_{t+L})$.
>
> 3. **Positive probability assumption:** We can meet this assumption by matching the marginal distribution of $x_t$ when the different values of $z_t$ are given.
>
> 4. **Sufficient changes:** The sufficient changes assumption is a technical condition on the latent variable density that pertains to the true data distribution and cannot be directly enforced in implementation. However, if this condition does not hold, the latent representation is generally not uniquely identifiable (because of linear dependence). In practice, this assumption can be partially tested or validated through empirical checks if necessary.
>
> In light of your comments, we have added the aforementioned discussion in the updated version. Thank you once again for your valuable input!
>
>
> >Q2: For the proof itself, the KL term would not vanish due to lack of invertibility induced by
>  (and even perhaps the multi-layer structure). There could be an argument for why this is still fine, but the proof is not there yet.
>
> A2: Thank you for the follow-up question. In the ELBO formulation, the KL divergence term measures the difference between the estimated variational distribution and the the true posterior. Importantly, the derivation of the ELBO does not explicitly rely on the relationship from latent variables to observed variables. Therefore, the KL divergence can be minimized to zero when the variational posterior is sufficiently expressive and the model is flexible enough to approximate the true posterior distribution.
>
> Thank you again for your constructive comments.

---

### Author Response · Authors · 2025-08-08
**Author Rebuttal by Authors**

Comment: Dear Reviewers and ACs,

We sincerely thank you for your time, effort, and thoughtful insights throughout both the initial review and discussion phases. It is encouraging that the reviewers think CHiLD is new, promising (Reviewer nYtx, 9SP8), novel (Reviewer nGAz and yDBF), and well-motivated (Reviewer yDBF). We here provide a general response to summarize the modifications of the paper.

- To Reviewer nYtx, we have provided a theoretical result of the consistency of estimation.
- To Reviewer nYtx, we have added experiments to show that more necessary information does not significantly affect the identification performance.
- To Reviewer nYtx, we have explicitly added a clarification in Section 5 and provided the compute resources in the updated version.
- To Reviewer 9SP8, we have revised Definitions 1 and 2 to explicitly state that $\hat{z}_t$ denotes the estimated latent variables inferred from observed variables.
- To Reviewer 9SP8, we have conducted additional experiments with 3 and 4 latent layers on five real-world datasets.
- To Reviewer nGAz, we have included a clear distinction of our contribution in the revised version.
- To Reviewer nGAz, we have highlighted the flexibility of our data generation process and layer dimensionality in the updated version.
- To Reviewer yDBF, we have highlighted the contribution of our work in the updated version.
- To Reviewer yDBF, we have provided a detailed explanation and the implications of the assumption in the updated version.
- To Reviewer yDBF, we have provided more simulation experiments with different types of nonlinear mixing functions.

Thanks again for your time dedicated to carefully reviewing this paper. We hope that our response properly addresses your concerns.

With best regards,

Authors of submission 8292

---

### Note · Authors · 2025-08-11

Dear Reviewers and AC,

We sincerely appreciate all of your constructive feedback, encouragement, and suggestions, especially during this busy period. We also thank you for your commitment to reviewing the paper directly and ensuring a fair evaluation. For the convenience of the final discussion, we summarize our responses to Reviewers **nYtx** and **9SP8** as follows:

---

**Reviewer nYtx**

Reviewer **nYtx** finds our work novel and promising for real-world applications. Before the rebuttal, the reviewer noted that a formal consistency result for the implementation should be provided, though this was not considered a major weakness.

During the rebuttal, we presented a theoretical result on the consistency of estimation and explained how the implementation aligns with our assumptions. We also added further experiments demonstrating that the availability of additional necessary information does not significantly affect identification performance.

After the rebuttal, the reviewer expressed satisfaction with our responses and retained their positive rating.

---

**Reviewer 9SP8**

Reviewer **9SP8** recognizes that our paper offers extensive theoretical contributions and promising experimental results.

Following the reviewer’s suggestions, we have refined key definitions and conducted additional experiments, which improved the readability and soundness of our work. The reviewer was satisfied with these additions and maintained a positive score supporting acceptance.

---

Thanks again for your time dedicated to carefully reviewing this paper. We hope that our response properly addresses your concerns. We would also like to sincerely thank the AC for their professional coordination and valuable guidance during the entire review process.

Best,

With best regards,

Authors of submission 8292

---

### Decision · Program_Chairs · 2025-09-17

**Decision:**

Accept (poster)

**Comment:**

## My review of the paper

This paper shows that hierarchical latent variables in temporal settings are identifiable by leveraging past, $x_{<t}$ and future $x_{>t}$ observations as context. They begin by showing blockwise identifiability of the L layers of hierarchical latents by assuming that one matches the joint over adjacent variables under an injective linear operator assumption. On the surface, block-identification of the layers doesn't tell you much---one hasn't yet disentangled the hierarchical relationship between latents---but this sets up their later results, and I thought that the linear operator assumption & proof technique was interesting. I've seen similar assumptions being used in the non-parametric instrumental variable literature (this connection is cited), but I'm not aware of prior work using these assumptions being used for causal representation learning. They then proceed to show that the sparsity of the hierarchical structure (where variables only depend on hierarchical parents and time lagged hierarchical) gives block identifiability of the latent layers, and finally conditional independence gives component-wise identifiability. These identifiability results are turned into an algorithm via a variational autoencoder that enforces the appropriate dependency structure. They evaluated their approach on both simulated data, where they achieved decent MCC results (though it is hard to say how well that would transfer to real settings), and a variety of real datasets. The real data experiments were impressively diverse (covering weather, stocks, human motion, etc.) but just focused on evaluating the observable variables (FID & correlation scores), with one qualitative experiment on the human motion dataset.

#### Strengths & weaknesses
 - I found the theory section very interesting - I was surprised at how much can be identified if you are prepared to assume injectivity of linear operators & the hierarchical structure. For this reason, I think the paper should be accepted.
 - While I appreciated the breadth of the experiments, I think that they would have been far more convincing if there was more effort put into trying to understand what hierarchical latent structure was estimated in practice. Causal representation learning has made a lot of theoretical progress, but the algorithms remain very difficult to apply on real problems because not enough effort has been put into this task of understanding the resulting latents (and critiquing the model).

Presentation issues & minor comments:

I felt that Section 2 / 3 could be improved by explicitly introducing a running example to illustrate the hierarchical data generating process that the authors have in mind. The section already uses examples from finance to illustrate the assumptions, so introducing that example earlier to provide examples of what you hope to identify & what makes it hard, would make these sections more accessible.

## Metareview

My opinion of the paper (captured in the review above) is fairly consistent with what the reviewers noted. I agree with Reviewer nYtx that the connection between the identifiability results and the algorithm should be formalized by including the outcomes of the discussion from the rebuttal in the text. There should also be a more explicit discussion on the limitations of applying these methods in practice (it is hard to know whether your assumptions are correct, and interpreting the latents is a challenge). But as I said in my review, I think the theoretical contributions are interesting, and the paper is worthy of acceptance.